



**Recent decrease trend of atmospheric mercury concentrations in East China: the influence of anthropogenic emissions**

Yi Tang[1,2], Shuxiao Wang[1,2*], Qingru Wu[1,2*], Kaiyun Liu[1,2], Long Wang[3], Shu Li[1], Wei Gao[4], Lei Zhang[5], Haotian Zheng[1,2], Zhijian Li[1], Jiming Hao[1,2]

[1] State Key Joint Laboratory of Environmental Simulation and Pollution Control, School of Environment, Tsinghua University, Beijing 100084, China

State Environmental Protection Key Laboratory of Sources and Control of Air Pollution Complex, Beijing 100084, China
School of Environment and Energy, South China University of Technology, Guangzhou, 510006, China
Yangtze River Delta Center for Environmental Meteorology Prediction and Warning, Shanghai, 20030, China
State Key Laboratory of Pollution Control & Resource Reuse, School of the Environment, Nanjing University, Nanjing, 210023, China

*Correspondence to:* Shuxiao Wang (shxwang@tsinghua.edu.cn)

       Qingru Wu (wuqingru06@163.com)

**Abstract**

Measurements of gaseous elemental Hg (GEM), other air pollutants including $SO_2$, $NO_X$, $O_3$, $PM_{2.5}$, CO, and meteorological conditions were carried out at Chongming Island in East China from March 1 in 2014 to December 31 in 2016. During the sampling period, GEM concentrations significantly decreased from $2.68 \pm 1.07$ ng m$^{-3}$ in 2014 to $1.60 \pm 0.56$ ng m$^{-3}$ in 2016. Monthly mean GEM concentrations showed a significant decrease with a rate of -0.60 ng m$^{-3}$ yr$^{-1}$ ($R^2$=0.6389, $p<0.01$ significance level). Combining the analysis of potential source contribution function (PSCF), principle component analysis (PCA), and emission inventory, we found that Yangtze River Delta (YRD) region was the dominant source region of atmospheric mercury in Chongming Island and





the main source industries included coal-fired power plants, coal-fired industrial boilers, and cement
clinker production. We further quantified the effect of emission change on the air Hg concentration
variations at Chongming Island through a coupled method of trajectory clusters and air Hg
concentrations. It was find that the reduction of domestic emissions was the main driver of GEM
decline in Chongming Island, accounting for 66% of the total decline. The results indicated that air
pollution control policies targeting $SO_2$, $NO_x$ and particulate matter reductions had significant co-
benefits on atmospheric mercury.



# 1 Introduction

Mercury (Hg) is of crucial concern to public health and the global environment for its neurotoxicity, long-distance transport, and bioaccumulation. The atmosphere is an important channel for global mercury transport. Once atmospheric Hg deposits to the aquatic system, it can be transformed into methylmercury (MeHg) which bio-accumulates through the food web and affects the central nervous system of human beings (Mason et al., 1995). Hg is therefore on the priority list of several international agreements and conventions dealing with environmental protection, including the *Minamata Convention on Mercury*.

Atmospheric Hg exists in three operationally defined forms: gaseous elemental mercury (GEM), gaseous oxidized mercury (GOM), and particulate-bound mercury (PBM). And the sum of GEM and GOM is known as total gaseous mercury (TGM). In the atmosphere, Hg mainly presents as GEM, accounting for over 95% of the total. GEM is stable in the troposphere with a long residence time (0.5 - 2 years) and can be transported at regional and global scale (Lindberg et al., 2007). GEM can be oxidized through photochemical reaction to GOM, which can be converted to PBM upon adsorption/absorption on aerosol surfaces. Both GOM and PBM are more soluble and quickly scavenged through dry and wet deposition (Schroeder and Munthe, 1998).

The atmospheric Hg observation results are important evidences to assess the effect of Hg emission control. During the past decades, significant decreases of GEM concentrations in Europe and North America have been observed (Cole et al., 2013; Weigelt et al., 2015). Air Hg concentrations in the northern hemisphere are reported to decline by 30-40% between 1990 and 2010 (Zhang Y et al., 2016). Such a decrease is consistent with the decrease in anthropogenic Hg emissions inventory in Europe and North America (Streets et al., 2011). So far, most of the long-term observations on the ground sites have been carried out in the developed countries. For the developing countries such as China, limited atmospheric Hg observations have been carried out (Fu et al., 2008b; Zhang H et al., 2016; Hong et al., 2016) and there is no official national monitoring network of atmospheric Hg. Therefore, there are few continuous multi-year observation records of China's air Hg published (Fu et al., 2015).

China is the largest emitter of atmospheric Hg in the world. Atmospheric Hg emissions in China



accounted for 27% of the global total in 2010 (UNEP, 2013), which led to high air Hg concentrations
in China. Therefore, atmospheric Hg observations in China are critical to understand the Hg cycling
at both regional and global scale. China's Mercury emissions had increased from 147 t yr$^{-1}$ in 1978
to around 538 t yr$^{-1}$ in 2010 due to the dramatic economic development (Zhang L et al., 2015; Wu
et al., 2016; Hui et al., 2017). Atmospheric mercury monitoring that spanned the longest periods
(from 2002 to 2010) in Guiyang, southwestern China witnessed the increase of mercury emissions
in China (Fu et al., 2011). However, recently atmospheric Hg emissions in China have been
estimated decreasing since 2012 (Wu et al., 2016). This decreasing trend needs to be confirmed by
atmospheric Hg observations.

In this study, we measured GEM, other air pollutants (eg., PM$_{2.5}$ and NO$_x$), and meteorological

parameters (eg., temperature and wind speed) at a remote marine site of Chongming Island in East
China during 2014-2016. We analyzed annual and seasonal variation of GEM and the potential
impact factors. Combining the analysis of potential source contribution function (PSCF), principle
component analysis (PCA), and emission inventory, the potential source regions and source
industries of atmospheric Hg pollution at the monitoring site are identified. In addition, a coupled
trajectories and air Hg concentration method is developed to assess the effect of Hg emission change
from different regions on air GEM concentration variation at the monitoring site.
## 2 Materials and methods
### 2.1 Site descriptions

The monitoring remote site (31°32′13″N, 121°58′04″E, about 10 m above sea level) locates at the

top of weather station in Dongtan Birds National Natural Reserve, Chongming Island, China (Figure
1). As China's third largest island, Chongming Island locates in the east of Yangtze River Delta
region with a typical subtropical monsoon climate. It is rainy, hot, with southern and southeastern
winds in summer and is dry, cold, and with northwestern wind in winter. The dominant surface types
are farmland and wetland. There are no large anthropogenic emission sources in the island and no
habitants within 5 km distance from the site. The downtown Shanghai area is 50 km to the southwest
of the site.



**2.2 Sampling methods and analysis**

During the monitoring period, we used Tekran$^{TM}$ 2537X/1130/1135 instruments to monitor speciated Hg in the atmosphere, which was widely used for air Hg observation in the world. Continuous 5-minute of GEM was measured by Tekran$^{TM}$ 2537X Hg vapor analyzer with the detection limit of 0.1 ng m$^{-3}$ at a sampling flow rate of 1.0 L min$^{-1}$ during two campaigns: March 1, 2014 to December 31, 2015 and March 26 to December 31, 2016. The sampling inlet was 1.5 m above the instrument platform.

The 2537X analyzer was calibrated automatically every 25 h using the internal Hg permeation source inside the instrument, and the internal permeation source was calibrated every 12 months with manual injection of Hg by a syringe from an external Hg source (module 2505). Two zero and two span calibrations were performed for each calibration of gold trap A and B, respectively. The error between gold trap A and gold trap B was limited to ±10 %. The impactor plates and quartz filter were changed in every two weeks. The quartz filter was changed once a month. The denuders were recoated once every two weeks following the procedure developed by Landis et al. (2002).

During the sampling campaigns, $PM_{2.5}$, $O_3$, $NO_x$, CO and $SO_2$ were also monitored by Thermo Scientific TEOM 1405D, Model 49i $O_3$ Analyzer, Model 48i CO Analyzer, Model 42i-TL NOx Analyzer and Model 43i $SO_2$ Analyzer, respectively. The detection limits of $O_3$, $SO_2$, $NO_x$, CO and $PM_{2.5}$ are 1.0, 0.5, 0.4, 0.04 and 0.1 μg m$^{-3}$, respectively. The meteorological parameters including air temperature, wind speed, and wind direction are measured by Vantage Pro2 weather station (Davis Instruments). The instruments are tested and calibrated periodically. All data are hourly averaged in this study.

**2.3 Sources apportionment of atmospheric mercury pollution**

2.3.1 PSCF model

To identify the source areas for pollutants with a relatively long lifetime such as GEM (Xu and Akhtar, 2010), the PSCF values for mean GEM concentrations in grid cells in a study domain are calculated by counting the trajectory segment endpoints that terminate within each cell. The number of endpoints that fall in the $ij$-th cell are designated $n_{ij}$. The number of endpoints for the same cell having arrival times at the monitoring site corresponding to GEM concentrations higher than a specific criterion is defined to be $m_{ij}$. The criterion in this study is set as the average Hg concentration



during our study period. The PSCF value for the *ij*-th cell is then defined as:
$$PSCF_{ij} = \frac{m_{ij}}{n_{ij}} W_{ij} \qquad (1)$$

where $W_{ij}$ is an empirical weight to reduce the effects of grid cells with small $n_{ij}$ values. In this
study, $W_{ij}$ is defined as in the following formula, in which $Avg$ is the mean $n_{ij}$ of all grid cells with
$n_{ij}$ greater than zero:
$$W_{ij} = \begin{cases} 1.0 & n_{ij} > 2*Avg \\ 0.7 & Avg < n_{ij} \le 2*Avg \\ 0.42 & 0.5*Avg < n_{ij} \le Avg \\ 0.17 & n_{ij} \le 0.5*Avg \end{cases} \qquad (2)$$

The PSCF value indicates the probability of a grid cell through which polluted events occurs.
More method details can be found in the study of Polissar et al. (Polissar et al., 1999). In this study,
the domain that covered the potential contribution source region (105°–135°E, 15°–45°N) was
divided into 22500 grid cells with 0.2°×0.2° resolution. 72-hour back trajectories were generated
hourly from 1 March, 2014 to 31 December, 2015 and from March 26 to December 31 in 2016 by
TrajStat, a software including HYSPLIT for trajectory calculation with trajectory statistics modules
(Wang et al., 2009). PSCF map was plotted using ArcGIS version 10.1.
2.3.2 Principal component analysis (PCA)
Correlation between Hg and other pollutant concentrations are used to identify source industries.
Strong positive loadings (loading>0.40) with $SO_2$ and $PM_{2.5}$ typically indicate the impact of coal
combustion, and strong positive loadings with GEM and CO have often been used as an indicator
for regional transport because both pollutants have similar source and stable chemical properties
(Lin et al., 2006; Pirrone et al., 1996). In this study, PCA was applied to infer the possible influencing
factors of GEM in 2014 and 2016. Prior to analysis, each variable was normalized by dividing its
mean, and pollutant concentrations ($SO_2$, CO, $NO_X$, $PM_{2.5}$) were averaged to 1-h sampling intervals
to match the hourly mercury monitoring during sampling period. The results in 2016 had no CO
data due to instrument broken. Statistics analyses were carried out by using SPSS 19.0 software.
**2.4 Quantification method of source contribution**
To further quantitatively assess the effect of change in emissions from different regions on air
concentrations variation at a certain monitoring site, a quantitative estimation method which coupled
trajectories with air Hg concentrations was developed. We firstly identified the trajectories by using



the National Oceanic and Atmospheric Administration (NOAA) Hybrid Single-Particle Lagrangian
Integrated Trajectory (HYSPLIT) model. The gridded meteorological data at a horizontal resolution
of $1°\times1°$ were obtained from the Global Data Assimilation System (GDAS) (Draxler and Hess,
1998). The starting heights were set to be 500 m above ground level to represent the center height
of boundary layer where pollutants are usually well mixed in boundary layer. Secondly, each
trajectory was assigned with GEM concentration by matching the arriving time in Chongming site.
Third, the backward trajectories which coupled with Hg concentrations were clustered into groups
according to transport patterns by using NOAA HYSPLIT 4.7. Thus, the grouped clusters were
applied to identify the Hg source regions. The Hg average concentration of the cluster $j$ was then
calculated as equation (3). And, the trajectory weighted concentration in the cluster $j$ as equation
(4). At last, the contribution of reduction at a certain region on Hg concentration at monitoring sites
in a certain period can be calculated as equation (5).

$$C_{j,t} = \frac{\sum_{i=1}^{n} C_{i,j,t}}{\sum_{i=1}^{n} N_{i,j,t}} \qquad (3)$$

$$TWC_{j,t} = \frac{\sum_{i=1}^{n} N_{i,j,t}}{\sum_{j=1}^{m} \sum_{i=1}^{n} N_{i,j,t}} \times C_{j,t} \qquad (4)$$


where $N$ refers to a certain trajectory. $j$ refers to a certain cluster. $t$ is the studied period, and $n$ is
the number of trajectory. $m$ is the number of cluster. $C$ is the GEM concentration, ng m$^{-3}$. $TWC$ refers
to the trajectory weighted concentration, ng m$^{-3}$

$$CR_j = \frac{TWC_{j,t_2} - TWC_{j,t_1}}{\sum_{j=1}^{m} TWC_{j,t_2} - \sum_{j=1}^{m} TWC_{j,t_1}} \qquad (5)$$

where $CR$ refers to the contribution of GEM reduction. $t_1$ and $t_2$ refers to the two period
participating to comparison, namely year 2014 and 2016 in this study, respectively.
This approach is a simple method to quantify the influence of anthropogenic emissions on GEM



concentration variation. It should be noted that errors always exist in calculating trajectories, causing
uncertainties in all trajectory-based approaches. Trajectory errors vary considerably in different
situation. Draxler (1996) suggested uncertainties might be 10% of the travel distance. Besides, this
method required similar meteorological conditions of the periods participated in comparison so as
to reduce the interference from meteorology.
**2.5 Regional atmospheric Hg emissions**
Regional atmospheric Hg emissions by month are calculated by using both the technology-based
emission factor methods and transformed normal distribution function method. Detailed
introduction of these two methods are described in our previous study (Wu et al., 2016).
Conventional air pollutant ($SO_2$, $PM_{2.5}$, and $NO_x$) emissions were calculated following the study of
Zhao et al. (2013). The source regions included in the emission inventory consisted of Shanghai,
Jiangsu, Zhejiang, and Anhui provinces according to the PSCF results (See section 3.3). The studied
emission sectors included coal-fired power plants, coal-fired industrial boilers, residential coal-
combustion, cement clinker production, iron and steel production, zinc smelting, lead smelting and
other small emission sectors (eg., municipal solid incineration, biomass incineration, copper
smelting, aluminum production, gold production, other coal combustion, oil combustion, and
cremation). The monthly Hg emissions were mainly distributed according to fuel combustions or
products productions by month (Table S1). For small emission sectors, the annual emissions were
equally distributed into monthly emissions. The GEM emissions from natural sources followed the
study of Wang et al. (2016).
**3 Results and discussions**
**3.1 Decreasing trends of atmospheric Hg during 2014-2016**
The average concentrations of GEM in 2014 and 2016 were 2.68±1.07 ng m$^{-3}$ and 1.60±0.56 ng
m$^{-3}$, respectively. The GEM concentrations in 2014 were significantly higher than the Northern
Hemisphere back-ground concentration (about 1.5 ng m$^{-3}$) (Sprovieri et al., 2010) and those
measured in other remote and rural locations in China (Zhang H et al., 2015; Fu et al., 2008a; Fu et
al., 2009). However, in 2016, the GEM concentration (1.60±0.56 ng m$^{-3}$) was similar to the
background concentrations in the Northern Hemisphere. During this period, monthly GEM





concentrations showed a significant decrease with a rate of -0.60 ng m$^{-3}$ yr$^{-1}$ (R$^2$=0.6389, p<0.01
significance level) (Figure 2).
Table 1 showed the Hg variation trends in different regions. Significant decreases of GEM
concentrations in North hemisphere over the past two decades have been well documented (Weigelt
et al., 2015; Cole et al., 2013; Kim et al., 2016). Weigelt et al. (2015) showed that GEM
concentrations decreased from 1.75 ng m$^{-3}$ in 1996 to 1.4 ng m$^{-3}$ in 2009 at Mace Head, Europe.
Ten-year trends of GEM concentrations at six ground-based sites in the Arctic and Canada also
showed a decreasing trend at a rate of 13-35 pg m$^{-3}$ y$^{-1}$ (Cole et al., 2013). In South Korea, the
observed GEM concentration also had significant decrease in recent years (Kim et al., 2016). In
south hemisphere, at the Cape Point of South Africa, GEM concentrations decreased from 1.35 ng
m$^{-3}$ in 1996 to 0.9 ng m$^{-3}$ in 2008 and rose after then (Martin et al., 2017). However, limited GEM
monitoring sites and relative short-time spans in China restricted the views of long-term trends in
atmospheric Hg concentration in this region. A preliminary assessment indicated that atmospheric
Hg concentrations in China kept increasing before 2012 (Fu et al., 2015). The decreasing trend
observed in our study was accordant with the unpublished data in Mt. Changbai during 2014-2015
cited in the review of Fu et al. (2015). But much sharper decrease of Hg concentrations was observed
in our study. The specific reasons for the Hg concentration decrease in our study will be discussed
in section 3.4. One potential worry is that the calculated trend will be sensitive to seasonal variation
and the missing data in January and February of both 2014 and 2016 may impact the downward
trend. To evaluate the impact of the missing data, we estimate the Hg concentrations in the missing
months based on the least squares method from the data of the same months during 2011-2017
(Figure S1). Combining the estimated data, we re-fit the Hg concentrations and downward trend
still maintained robust and similar to the downward trend in manuscript (Figure 2 and Figure S2).
Thus, we assume that the missing data is not very important and will not impact our main conclusion.
**3.2 Seasonal variation of GEM concentrations**
Figure 3 showed the monthly variation of GEM concentrations in Chongming Island during the
monitoring period. Observed GEM concentrations showed an obvious seasonal cycle. The mean
GEM concentration in warm season (from April to September) is 0.29 ng m$^{-3}$ higher than that in
cold season. Such seasonal variation trend is also observed at Nanjing, Miyun, Mt. Ailao, Mt.





Waliguan, and Shangri-La (Zhang et al., 2013; Zhang et al., 2016; Fu et al., 2015; Zhu et al., 2012).
On the other hand, the means of GEM at Mt. Gongga, Mt. Daimei, Mt.Leigong, and Mt. Changbai
in China are relatively higher in cold seasons. The average of atmospheric Hg concentrations in the
north hemisphere also have a trough value in summer (Sprovieri et al., 2016).

Seasonal variations of GEM concentration are generally attributed to the following factors,

including natural and anthropogenic emissions, atmospheric chemical reaction, and air mass
transportation. The higher Hg concentrations in cold seasons in Mt. Ailao and Mt. Waliguan were
mainly explained by coal-combustion for urban and residential heating during cold seasons.
Whereas, increasing solar radiation and soil/air temperature dominate the higher Hg concentrations
in Mt. Gongga and Mt. Leigong. In addition, sites in southern, eastern, and northeastern China also
impacted from anthropogenic emissions of GEM from the north and west by the northerly winter
monsoon while the sites located in western, southwestern, and northern China were impacted in
the warm season (Fu et al., 2015). As to most sites in the northern hemisphere, high wet Hg
precipitation induced probably by faster GEM oxidation led to lower Hg concentrations in summer.

As to the observation site in Chonming island, we observed almost synchronized trends

between emissions and air Hg concentrations in Figure 4. The annual emissions from both natural
source and anthropogenic source in YRD region (Anhui, Zhejiang, Jiangsu, and Shanghai) was -
0.75 and 10.3 t, respectively. It should be pointed that the natural emissions here is a net natural
emissions, which is the byproducts of a bi-directional Hg flux, time, and area. When the data is
negative, it means GEM dry deposition to the calculated surfaces. Otherwise, it means GEM
emissions to air. The natural emissions varied from -5.4 t to 8.4 t with the highest value in summer
and the lowest value in winter. The anthropogenic emissions were in the range of 2.5-2.7 t, which
is almost unchanged compared to the natural emissions. Therefore, we supposed that the seasonal
cycle of GEM concentrations was dominated by natural emissions (Figure 4). The seasonal trend
of natural emissions is closely related with the canopy types in YRD areas, where widely
subtropical forests, paddy field, and dry farming were observed (Figure S3). The high temperature
will speed up decomposition of organic compound in soil, which lead to Hg emissions from
farmland and forest in YRD region (Luo et al., 2016; Yu et al., 2017). In autumn and winter, with
the decrease of temperature (Table S2), the role of soil changed from Hg source to sink, which



reduces the Hg concentrations in the air (Wang et al., 2016). At the same time, the growing
vegetation in autumn also absorbs air Hg, resulting lower Hg concentrations compared to that in
winter. Besides, more air mass transportation from North China and YRD was another reason of
higher Hg concentration in winter than that in autumn. According to the statistics of backward
trajectories in section 3.4, the air mass from North China and YRD region (NW and SW in Table
S3) in autumn and winter accounted for 73% and 95% of the total trajectory in autumn and winter,
respectively. We also noted that the seasonal variation of emissions is more significant than that of
Hg concentrations. Higher wet Hg deposition in summer is a potential impact factor, which reached
about 6.6 times of that in winter (Zhang et al., 2010). On one aspect, abundant Br at the coastal
site of Chongming and higher $O_3$ concentrations and solar radiation will lead to faster GEM
oxidation in summer.

## 3.3 Source apportionment of atmospheric Hg pollutions

According to the PSCF result, YRD region, including Shanghai, Jiangsu, Anhui, and Zhejiang
provinces, was the dominant source region in both 2014 and 2016 (Figure 5). Therefore, Hg
emissions from these areas would contribute to high proportion of Hg pollution in Chongming
Island. The offshore area mainly around Jiangsu province also has a high PSCF value because some
trajectories from North China, especially Shandong province, transport to Chongming Island
through this area. Compared to the result in 2014, the PSCF value had an obvious decline in East
China Sea in 2016. This decline may be contributed by the downward trend of GEM concentrations
in north hemisphere (Zhang et al., 2016).
PCA method was applied to preliminarily identify the source industries. In the studied period,
totally 2 factors were identified in 2014 and 2016, respectively. The factor 1 had strong factor
loadings of GEM, $SO_2$, $NO_x$, CO, and $PM_{2.5}$ in both 2014 and 2016 (No CO data in 2016 due to
equipment problems). The factor 1 accounted for 49% variance in 2014 and 50% variance in 2016
(Table 2). The results indicated common significant source sectors of the above five air pollutants,
which can also be proven from emission inventories (Table 3). The dominant source industries
included coal-fired power plants, coal-fired industrial boilers, and cement clinker production. The
PCA results showed that anthropogenic emissions were the main sources of GEM during the
sampling period.



The factor 2 in 2014 and 2016 both had a strong positive loading on $O_3$ and negative loading on
$NO_x$. The anti-correlation between $O_3$ and its precursor $NO_x$ could be an indication of air exchange
between planet boundary layer (PBL) and troposphere. However, the low loading on GEM of factor
2 indicated that Factor 2 had no relationship with GEM concentrations at Chongming from the
aspect of whole year data.
**3.4 The influence of anthropogenic emissions**
To further understand the reason of the downward trend, we firstly compared the meteorological
conditions in both 2014 and 2016. We noted that the difference of annual temperature, solar radiation,
and relative humidity were constrained in the range of 17.13±7.48 °C, 165.55±45.87 W m$^{-2}$ and
75.38±5.82%, respectively (Table S2). The coefficient of variation for annual mean of these
meteorological conditions in 2014 and 2016 was 2.6%, 6.7% and 0.2%, respectively. In addition,
the wind rose was similar, and the dominating wind was from SE in both 2014 and 2016 (Figure
S4). The HYSPLIT results also provided similar trajectories in 2014 and 2016 (Figure 6). Therefore,
we assumed that the meteorological condition was not the dominant reason of GEM decline at
Chongming site.
To further quantify the driver of GEM decline, a trajectory-based analysis method was used in
this study. The 72-h air mass back trajectories were calculated using HYSPLIT for every 8 hours
starting at the observation site. Approximately 918 and 832 trajectories were calculated in sampling
period in 2014 (Mar 1 to Dec 31, 2014) and 2016 (Mar 26 to Dec 31, 2016), respectively. The
trajectories were grouped into 3 clusters in each year according to geographical regions (Figure 6).
The first cluster of trajectories mainly passed through the regions (eg., North China) north and
northwest to Chongming Island before arriving to our monitoring site, which was denoted as cluster
NW. The second cluster mainly passed through the regions west and southwest to Chongming,
which was signed as cluster SW. The third type mainly originated from the East China Seas, South
Korea, Japan and Northeast Asia continent, and then arrived to our monitoring sites directly without
passing the mainland China. This type of trajectories was named as cluster EAST. Some trajectories
originated from the East China Sea and crossed the mainland China before arriving Chongming
were grouped into cluster NW or SW depending on the regions it crossed. The trajectories for each
of the three clusters in 2014 and 2016 were shown in Table 4.



Table 4 showed the detail statistics data of the three classifications. From 2014 to 2016, the whole
China region (NW, SW) contributed to 66% of GEM decline at Chongming Island. These results
reflected the effectiveness of existing air pollution control measures in China (SC, 2013; MEP, 2014).
Meanwhile, the NW region, SW region, and EAST region causes 47%, 19%, and 34% for GEM
decline, respectively (Table 4). The largest contribution of reduction was observed in the cluster
NW, suggesting that air pollution controls on anthropogenic emissions in NW region dominated the
recent decrease of GEM concentrations at Chongming site. We also noted that the largest decline of
Hg concentrations was observed in the cluster SW (1.51 ng m$^{-3}$), which indicated more effective air
pollution control in the regions where the air mass of the cluster SW passed. However, since the
proportion of the trajectories in the cluster SW was much less than that in the cluster NW, the
contribution of cluster SW to GEM decline in our observation site was lower.
**4 Conclusion**
Atmospheric Hg was continuously measured for three years at a regional background site in the
YRD region. During the sampling period, a downward trend for GEM concentrations (-0.60 ng m$^{-3}$
y$^{-1}$) at Chongming Island was observed. The seasonal GEM cycle was dominated by the natural
emissions while the annual GEM concentration trend was mainly impacted by anthropogenic
emissions. By using a new approach that considers both cluster frequency and the Hg concentration
associated with each cluster, we quantified that atmospheric Hg from NW region, SW region, and
EAST region have caused 47%, 19%, and 35% decline of GEM concentrations at Chongming
monitoring site, respectively. The result suggested that reduction of anthropogenic emissions in
mainland China was the main cause of the recent decreasing trend of GEM concentration at
Chongming site. The air pollution control policies in China, especially the pollution control in the
coal-fired power plants, coal-fired industrial boilers, and cement clinker production in YRD region
and Shandong province, have received significant co-benefit of atmospheric mercury emission
reductions. On the other hand, emission reduction from the EAST region, where clusters arrived to
Chongming monitoring site directly without passing the mainland China, implies global effort on
atmospheric Hg emission control under the guidance of *Minamata Convention on Mercury*.
Considering that the *Minamata Convention on Mercury* had come into force in 2017, continuous





long-term observation of atmospheric Hg in China will be required for the assessment of policy
effectiveness.

*Data availability.* All data are available from the authors upon request.

*Competing interests*. The authors declare that they have no conflict of interest.

*Acknowledge.* This work is sponsored by the Natural Science Foundation of China (No. 21607090),
Major State Basic Research Development Program of China (973 Program) (No. 2013CB430000),
National Key R&D Program of China (No. 2016YFC0201900)









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



**Figure citation**
**Figure 1.** The location of the Chongming monitoring site in Shanghai, China
**Figure 2.** Trends of monthly average GEM concentrations and their least squares fit
**Figure 3.** Monthly variations of GEM concentration at remote sites in China (Fu et al.,
2015;Sprovieri et al., 2016)
**Figure 4.** Seasonal cycle of GEM concentrations and emissions during 2014-2016. The bars
represent the standard deviation of seasonal average.
**Figure 5.** Source regions of GEM at monitoring site from PSCF model in 2014(a) and 2016(b)
**Figure 6.** The back trajectories map for cluster NW, SW and EAST in 2014(a) and 2016(b)
(NW – Northwest; SW – Southwest; EAST – East).





**Table citation**
**Table 1.** Historical variation trends of atmospheric Hg in previous studies
**Table 2.** PCA component loading of GEM and the co-pollutant**s**
**Table 3.** Main air pollutant emitted by the different sector in YRD region
**Table 4.** The statistics of cluster and estimated contribution of GEM reduction in 2014 and 2016






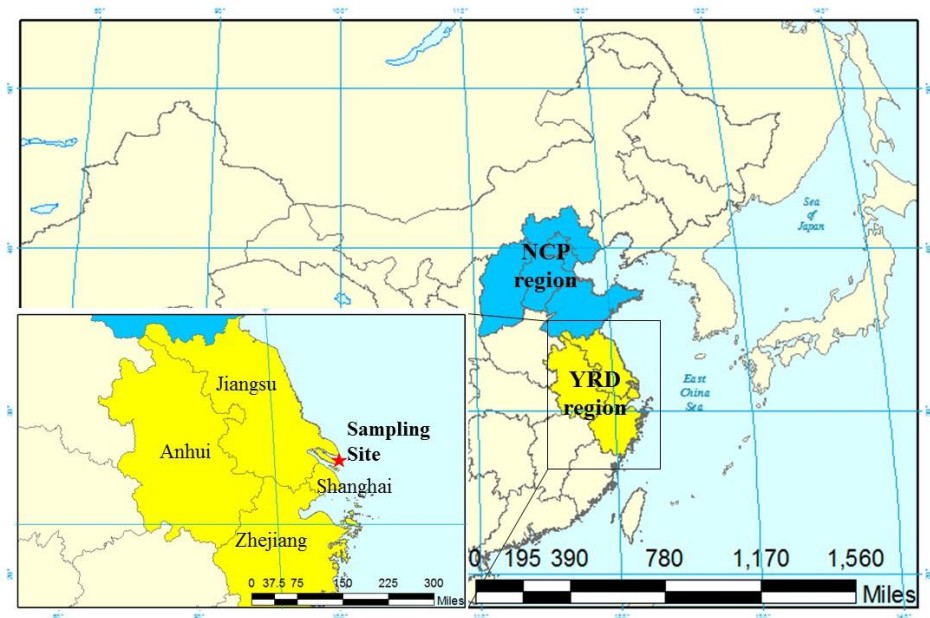

**Figure 1**. The location of the Chongming monitoring site in Shanghai, China



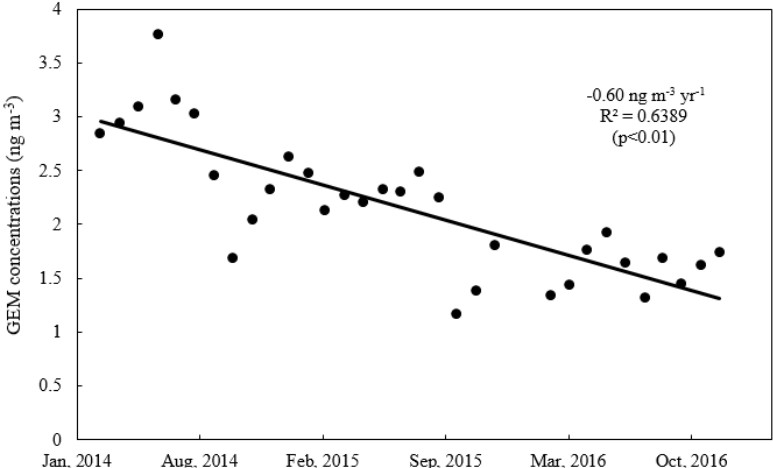

**Figure 2.** Trends of monthly average GEM concentrations and their least squares fit





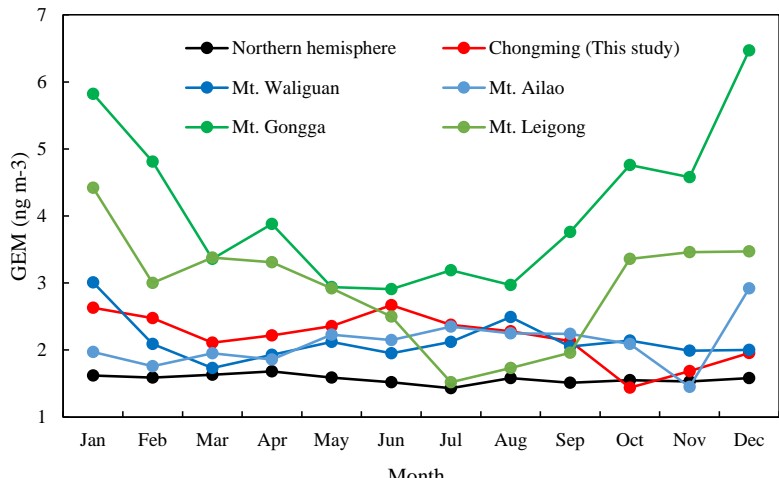

**Figure 3.** Monthly variations of GEM concentration at remote sites in China (Fu et al.,
2015;Sprovieri et al., 2016)



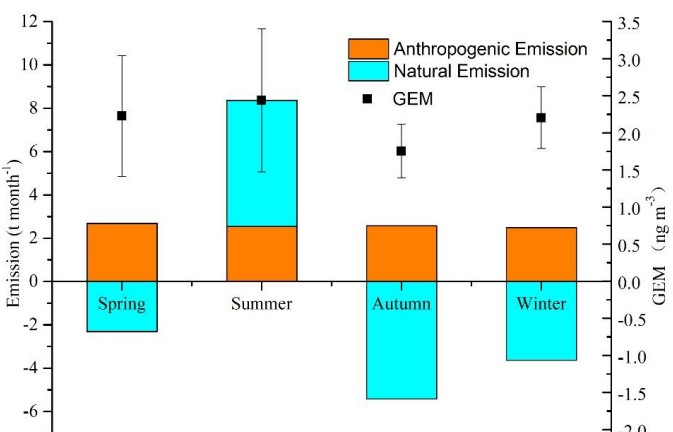


**Figure 4.** Seasonal cycle of GEM concentrations and emissions during 2014-2016. The bars

represent the standard deviation of seasonal average.






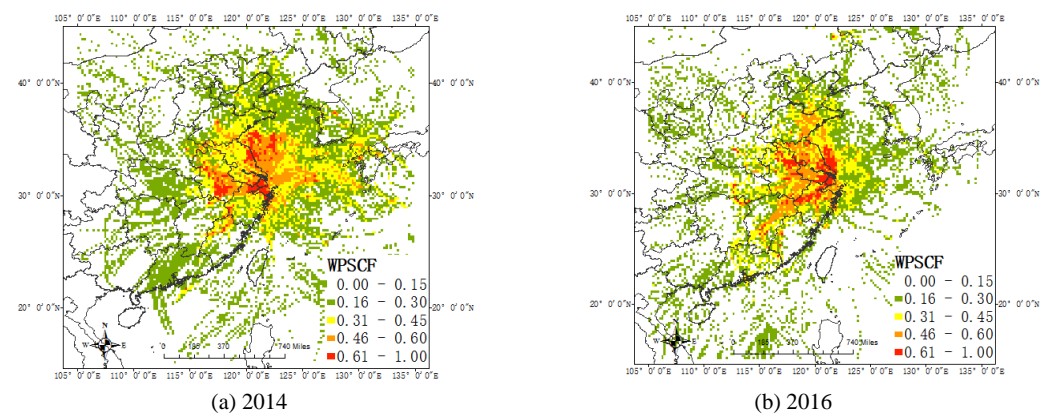

          (a) 2014                                            (b) 2016

**Figure 5.** Source regions of GEM at monitoring site from PSCF model in 2014(a) and 2016(b)





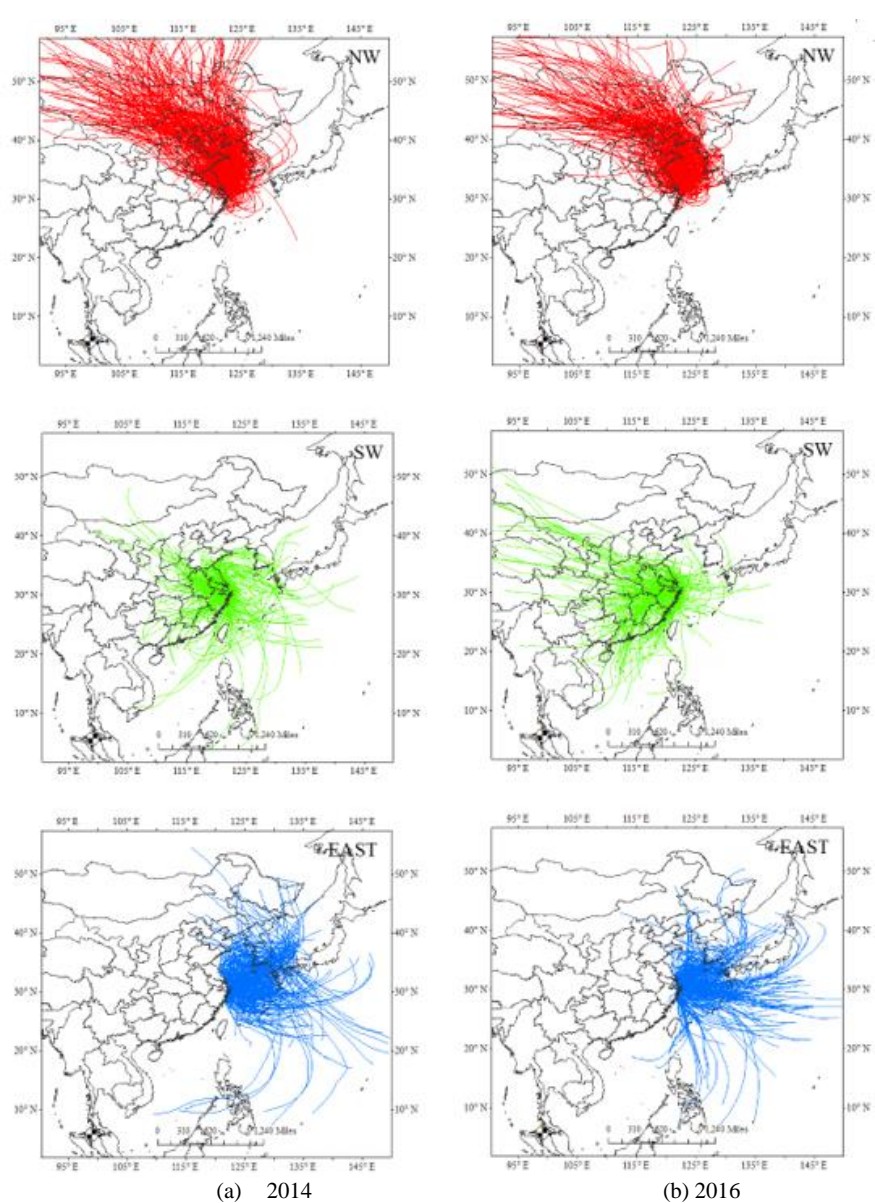

(a) 2014                                    (b) 2016

**Figure 6.** The back trajectories map for cluster NW, SW and EAST in 2014(a) and 2016(b)

(NW – Northwest; SW – Southwest; EAST – East)





**Table 1.** Historical variation trends of atmospheric Hg in previous studies

| Monitoring site | Duration | TGM trend (pg m$^{-3}$ yr$^{-1}$) | Variation trend | Site description | References |
|---|---|---|---|---|---|
| Alert, Canada | 2000-2009 | -13(-21,0) | -0.9%/y | remote arctic, tundra | Cole et al. 2013 |
| Kuujjuarapik, Canada | 2000-2009 | -33(-50,-18) | -2.1%/y | forest/tundra, sub-arctic | Cole et al. 2013 |
| Egbert, Canada | 2000-2009 | -35(-44,-27) | -2.2%/y | forest/agricultural, urban within 100km | Cole et al. 2013 |
| Zeppelin Stn, Norway | 2000-2009 | +2(-7,+12) | no trend | remote arctic mountain ridge, tundra | Cole et al. 2013 |
| St.Anicet, Canada | 2000-2009 | -29(-31,-27) | -1.9%/y | flat, grassy, rural, urban/industrial within 100 km | Cole et al. 2013 |
| Kejimkujik, Canada | 2000-2009 | -23(-33,-13) | -1.6%/y | Forested rural | Cole et al. 2013 |
| Head, Ireland | 1996-2009 | | -1.3±0.2%/y | located on the western coast of Ireland, the nearest city is 55km to the east | Weigelt et al. 2015 |
| Hannam dong, South Korea | 2004-2011 | no trend (3.54±1.46 ng/m3) | | In the center of Seoul metropolitan city | Kim et al. 2016 |
| Hannam dong, South Korea | 2013-2014 | decrease to 2.34±0.73 ng/m3 | | | Kim et al. 2016 |
| Chongming Island, China | 2014-2016 | -520 | -29.4%/y | remote, wet land and farmland with 20km | This study |









**Table 2**. PCA component loading of GEM and the co-pollutants

| | 2014 | | | 2016 | |
|---|---|---|---|---|---|
| | Factor 1 | Factor 2 | | Factor 1 | Factor 2 |
| $SO_2$ | **0.763** | 0.142 | $SO_2$ | **0.821** | -0.088 |
| $NO_X$ | **0.766** | -0.201 | $NO_X$ | **0.699** | **-0.522** |
| $O_3$ | -0.113 | **0.977** | $O_3$ | -0.41 | **0.968** |
| $PM_{2.5}$ | **0.853** | 0.052 | $PM_{2.5}$ | **0.875** | 0.053 |
| GEM | **0.664** | 0.024 | GEM | **0.775** | -0.19 |
| CO | **0.793** | 0.12 | | | |
| Component | Combustion | Exchange of PBL and troposphere | Component | Combustion | Exchange of PBL and troposphere |
| Variance explain | 49.359 | 17.525 | Variance explain | 50.625 | 75.735 |

Note: Text in bold phase were regarded as high loading (factor loading>0.40)





**Table 3.** Main air pollutant emitted from the different sectors in YRD region

|  | $SO_2$ (kt) | $NO_x$(kt) | $PM_{2.5}$(kt) | GEM(t) |
|---|---|---|---|---|
| Coal-fired power plants | 1823.76 | 1638.31 | 171.31 | 18.52 |
| Coal-fired industrial boilers | 1526.73 | 649.94 | 198.11 | 17.02 |
| Residential coal combustion | 344.57 | 77.34 | 95.21 | 1.04 |
| Cement clinker production | 421.17 | 622.33 | 362.27 | 11.56 |
| Iron and steel production | 505.03 | 156.16 | 240.6 | 3.99 |
| Biomass incineration | 13.99 | 92.25 | 556.18 | 0.61 |
| Other sectors | 7225.32 | 2872.43 | 832.05 | 5.39 |







**Table 4.** The statistics of cluster and estimated contribution of GEM reduction in 2014 and 2016

| Year | Cluster | Trajectories | | GEM concentration, $C_j$ (ng m$^{-3}$) | Trajectory weighted concentration, $TWC_j$, (ng m$^{-3}$) | Contribution to GEM reduction , $CR_i$ |
|------|---------|---------|-------|------|------|------|
| | | Numbers | Ratio | | | |
| **2014** | NW | 414 | 45% | 2.46 | 1.11 | |
| | SW | 215 | 23% | 3.37 | 0.79 | |
| | EAST | 289 | 31% | 2.58 | 0.81 | |
| **2016** | NW | 322 | 39% | 1.56 | 0.49 | 47% |
| | SW | 258 | 31% | 1.86 | 0.62 | 19% |
| | EAST | 252 | 30% | 1.44 | 0.35 | 35% |
