# Peer review of "Recent decrease trend of atmospheric mercury concentrations in East China: the 1 2 influence of anthropogenic emissions Yi Tang1, 2, Shuxiao Wang1, 2\*, Qingru Wu1, 2\*, Kaiyun Liu1, 2, Long Wang3, Shu Li1, Wei Gao4, Lei 3 Zhang5, Haotian Zheng1, 2, Zhijian Li1, Jiming Hao1, 2 4 5 1 State Key Joint Laboratory of Environmental Simulation and Pollution Control, School of 6 7 Environment, Tsinghua Un"

_Atmospheric Chemistry and Physics, 2017_

## Referee Comment (RC1) · Anonymous Referee #1 · 16 Feb 2018

The paper by Tang et al. presents 33 months long measurements of GEM, SO2, NOx, O3, CO, and PM2.5 at an island not far from Shanghai. The authors find a pronounced GEM downward trend and analyse it in terms of regional sources, clusters of backward trajectories and principle component analysis. Seasonal variation is also discussed. The authors arrive at the conclusion that the downward trend is due to substantial reduction of the regional mercury sources. This is an important finding which deserves to be published. I think, however, that several substantial changes need to be made to the manuscript before it can be accepted for publication.

The suggestions are:

[Figure]

Experimental: The authors mention that mercury species were measured but only GEM data are presented and their trend calculated. What was the contribution of GOM and PBM? Could they contribute to the trend? This is important to discuss because the regional atmospheric Hg emissions in Section 2.5 are probably not only those of GEM but of total mercury. What was the seasonal variation of GOM and PBM? Could it provide some additional evidence for the reasons of GEM seasonal variation?

Section 3.1: Averages and their standard deviations should always be stated with the number of measurements because only then statistical tests for significance of differences can be made. In line 194 the authors claim that GEM concentrations in 2014 were significantly higher than. . . - at which significance level? Line 199: the annual decrease rate should be given with its uncertainty and number of months.

Figure 3: It is not clear how the points in Figure 3 were calculated? Were the data detrended before the averaging? In view of the strong downward trend they should be. What is the standard deviation or standard error of the monthly means – please show them as vertical bars in Fig. 3. Are the differences between the months statistically significant? This is a precondition for the discussion of the seasonal variation.

Figure 4 and its capture: This figure needs substantial revision to illustrate the point the authors make and to make it understandable for the readers. Negative emissions are deposition fluxes and should be named as such. Thus "natural emissions" in spring, autumn and winter are in fact deposition fluxes. Net fluxes are needed to illustrate the point made by the authors but they are not shown. The capture should also state that it is about the emissions and depositions in the YRD region? This will rise a problem: trajectory analysis in section 3.4 shows large influence of transport from the NW provinces of Chine outside of the YRD region. How does this transport influence the seasonal variation?

Line 243-245: "The annual emissions from both natural source and anthropogenic source . . .. was 0.75 and 10.3 t, respectively" – the reader may think that natural emissions make less than 10% of the anthropogenic ones and cannot thus be responsible for the seasonal variation. One has to look in Fig. 4 to find out that the "natural emissions" are a sum of natural emissions in one season and "natural" deposition fluxes in three seasons. That provokes a question: how is anthropogenically emitted mercury removed from the atmosphere if there are only "natural" deposition fluxes? Please use the correct terminology and separate the natural and anthropogenic emissions from the deposition fluxes of both.

The results of PCA analysis and Table 2: The authors attribute the factor 2 to "exchange of PBL with free troposphere" but do not explain why. Last row in the table 2 called "variance explain" lists for 2016 exchange of PBL with the free troposphere 75.735 which together with 50.625 for "combustion" makes more than 100. As such the units of "variance explain" cannot be percent. What are the numbers in this row and does it make sense to present them with three valid numbers after decimal point?

Table 3: The numbers are probably annual emissions but the capture does not say it. The year of the emissions is not given. I wonder about the "other SO2 sources" which are substantially larger than all coal, oil, and biomass burning taken together. If it is not an error, what are the "other SO2 sources"?

Chapter 3.4 misses a major point: Table 3 of SO2, NOx, PM2.5, and GEM emissions is only for one undefined year and only for the YRD region. Table 4 and Figure 6 show a dominant influence of transport from NW of China which is mostly outside of the YRD region. To illustrate convincingly the major conclusion of the paper one would need a table with the inventories for NW and SW (perhaps separately) and for 2014 and 2016.

Table 1: The paper is about regional trend and I wonder why it is necessary to discuss global background trends in such detail. Also because the reasons of downward trend of mercury at many background stations of the world at the time of increasing global emissions is still not well understood (compare Horowitz et al., EST 48, 10242-10250, 2014, with Soerensen et al., GRL, 39, L21810, doi:10.1029/2012GL053736, 2012).

Figure S1: Here the least square fit of 4 points provides R2 of 0.487 for January and 0.613 for February for which the authors claim p < 0.01 in both cases. This is surely incorrect because 4 points result statistically in only 2 degrees of freedom. Please explain.

Figure S2: The downward annual rate should be given with its standard error.

Editorial remarks:

Line 51: "Both GOM and PBM are more soluble.." than what? PBM is not necessarily more soluble than GEM but it is scavenged by wet deposition. Low solubility of GEM need to be mentioned before this statement.

Lines 71/72: ...have been estimated to decrease..

Line 86: ...is located...

Lines 102/103: ... the error between gold trap A and gold trap B was limited to...? Probably the difference instead of error was limited. What happens if the difference is more than the limit?

Lines 171/172: "uncertainties" would be better than "errors"

Line 199: Please state the decrease rate with its standard error.

Lines 207-209: A reference to Martin et al (2017) is not correct because the paper does not contain annual averages and the authors of this paper do not mention a gap in the measurements between 2004 and 2007. The correct reference would be: annual average GEM concentration decreased from 1.29 ng m-3 in 1996 to 1.19 ng m-3 in 2004 (Slemr et al., GRL 35, L11807, doi:10.1029/2008GL033741, 2008) and were increasing from 0.93 ng m-3 in 2007 (Slemr et al., ACP 15, 3125-3133, 2015) until 2016 (Martin et al, 2017).

---

## Referee Comment (RC2) · Anonymous Referee #2 · 20 Feb 2018

General comments

This paper presents a multi-year record of GEM concentrations at Chongming Island, East China and reports a decreasing trend with a rate of -0.52 ng/m3/yr. The authors attribute this decreasing trend to air pollution control policies targeting SO2, NOx, and particulate matter. This paper could make a valuable addition to the literature. However, while I agree with the conclusions (e.g., decreasing anthropogenic emissions, co-benefit from pollution control policies targeting other compounds), I am not really convinced by the level of scientific evidence presented here. The paper could be suitable for publication in ACP after the authors address the following issues.

[Figure]

**Major comments**

I really think you should perform a trend decomposition of the signal (signal = seasonal + trend + random, example here: https://anomaly.io/seasonal-trend-decomposition-in-r/). There is a very strong seasonal cycle and you conclude that "the seasonal GEM cycle was dominated by the natural emissions". However, how can you explain that the seasonality is way more pronounced in 2014? To me, presenting emissions inventories is not convincing enough; how you can you be sure that the decreasing trend is not driven by a change in seasonality? While SO2, NO2, and PM concentrations were monitored, data are not presented nor discussed. Do you also observe a decreasing trend? That would be the best way to support that "air pollution control policies targeting SO2, NO2, and PM reductions had significant co-benefits on atmospheric Hg". Finally, I wonder why GOM and PBM data are not reported and discussed. Do you also observe a decreasing trend? You may have encountered issues with the speciation unit. If so, was the experimental setup identical in 2014 and 2016, or did you analyze GEM when the speciation unit was working vs. TGM when it wasn't? A discussion on analytical uncertainties would be much welcomed.

**Line by line comments**

Line 26: "GEM concentrations showed a significant decrease with a rate of -0.60 ng/m3/yr". According to Table 1, the rate is -0.52 ng/m3/yr.

Line 33: "It was find" should be "It was found".

Lines 47-48: "In the atmosphere, Hg mainly presents as GEM, accounting for over 95% or the total". Can you please add a reference? Is that also true at your site?

Line 61-62: "(...) there is no official national monitoring network of atmospheric Hg". Out of curiosity, what is the current status of the Asian-Pacific Mercury Monitoring Network (http://nadp.sws.uiuc.edu/newIssues/asia/)? Do you think that Chinese sites will be included?

Lines 64-67: "Atmospheric Hg emissions in China accounted for 27% of the global total in 2010 (UNEP, 2013), which led to high air Hg concentrations in China. Therefore, atmospheric Hg observations in China are critical to understand the Hg cycling at both regional and global scale". Please define "high" air Hg concentrations. Additionally, in order to emphasize the fact that observations in China are critical to understand the Hg cycling on a global scale, you could perhaps add a sentence about 1) future projections (e.g., Chen et al., 2018; Pacyna et al., 2016), and 2) long-range transport of Chinese emissions to other regions (e.g., Chen et al., 2018; Corbitt et al., 2011; Sung et al., 2018).

Lines 93-94: "we used Tekran 2537X/1130/1135 instruments to monitor speciated Hg in the atmosphere". I wonder why GOM and PBM concentrations are not reported in the manuscript. If concentrations were recorded, it would be interesting to discuss the results. Do you also see a decreasing trend from 2014 to 2016? From 1978 to 2014, the fractions of GEM and PBM decreased, while the GOM emission share gradually increased (Wu et al., 2016). What about the speciation of emissions since 2014? Can you observe a trend in GOM/PBM concentrations? Alternately, did you have issues with the speciation unit? It is quite common and I would appreciate an open discussion about that and associated analytical uncertainties. What kind of issues did you encounter? Are you confident that you collected and analyzed GEM (vs. TGM) during the entire experiment? Was the instrumental setup exactly the same during the entire experiment? If not, how can you compare GEM concentrations without discussing analytical uncertainties? See major comment.

Lines 103-104: "The impactor plates and quartz filter were changed in every two weeks. The quartz filter was changed once a month". Did you change the quartz filter every two weeks or once a month?

Line 106: "During the sampling campaigns, PM2.5, O3, NOx, CO and SO2 were monitored". Why aren't you discussing the data, especially SO2, NOx, PM2.5 while your main conclusion is that Hg decreasing trend in due to air pollution control policies targeting SO2, NOx, and PM. I agree that you present emissions inventories, but I would really appreciate to see a real interpretation and discussion of these data. Do you also observe a decreasing trend? See major comment.

Lines 173-175: "Besides, this method required similar meteorological conditions of the periods participated in comparison so as to reduce the interference from meteorology". I am not sure I understand this sentence. Do you mean that you used similar meteorological data in 2014 and 2016 to compute the back-trajectories? Or are you referring to the fact that meteorological conditions were pretty much similar in 2014 and 2016 (lines 266-274)?

Lines 188: "For small emission sectors (. . .)". Which ones?

Lines 193-194: "The average concentrations of GEM in 2014 and 2016 were (. . .)". What about the mean concentration in 2015? Additionally, are the average annual concentrations actually referring to March-December? If so, please add something like "The average concentrations of GEM in 2014 (Mar-Dec) and 2016 (Mar-Dec) were (. . .)".

Lines 194-195: How does it compare to concentrations reported in Sprovieri et al. (2016)?

Lines 199-200: "During this period, monthly GEM concentrations showed a significant decrease with a rate of -0.60 ng/m3/yr". Table 1 refers to TGM concentrations, not GEM. Additionally, as mentioned earlier, the rate is -0.52 ng/m3/yr in Table 1. Please, try to be consistent throughout the manuscript.

Lines 201-216: To me, "GEM" and "TGM" are not interchangeable (see previous comment). While the difference between TGM and GEM is usually smaller than 1% (Soerensen et al., 2010), it might not be the case everywhere. What is the fraction of GOM at your site? I would appreciate a discussion on analytical uncertainties and instrumental setups. The sentence "at the Cape Point of South Africa, GEM concentrations

decreased from 1.35 ng/m3 in 1996 to 0.9 ng/m3 in 2008" is not entirely true. A downward trend has been observed from 1996 to 2005, while an upward one is observed since 2007 (Martin et al., 2017; Slemr et al., 2015). Additionally, the instrumental setup changed: a manual amalgamation technique was used from 1995 to 2004 while a Tekran instrument has been used since 2007 (Martin et al., 2017). It might also be the case at other stations in Table 1. How does it influence the various trends reported in Table 1?

Lines 212-214: "The decreasing trend observed in our study was accordant with the unpublished data in Mt. Changbai during 2014-2015 cited in the review of Fu et al. (2015). But much sharper decrease of Hg concentrations was observed in our study". Aren't the data at Mt. Changbai you are referring to in Sprovieri et al. (2016)? What is the trend at that site? Why isn't included in Table 1?

Line 224: Are you referring to Figure 2?

Lines 225-227: Is that based on the ∼3 years of data?

Section 3.2: I find this section quite confusing and difficult to follow.

Line 234: "The higher Hg concentrations in cold seasons in Mt. Ailao and Mt. Waliguan (. . .)". You say above that concentrations are lower in the cold season at these sites. This is confusing.

Line 250-251: "Therefore, we supposed that the seasonal cycle of GEM concentrations was dominated by natural emissions". How can you explain that the seasonal cycle is more pronounced in 2014 than in 2016? See major comment.

Lines 275-276: "This decline may be contributed by the downward trend of GEM concentrations in north hemisphere". Please, elaborate on this idea. I don't really understand what you mean here.

Section 3.4: See major comment. Please perform a trend decomposition of the signal. I don't know which software you use, but here is an example using R:

https://anomaly.io/seasonal-trend-decomposition-in-r/.

Lines 315-325: Do you get the same results if you perform this analysis on SO2, NOx, and PM concentrations?

Line 318: 34% should be 35% according to Table 4. Additionally, how can you explain this result? Is there a decline in anthropogenic emissions and a GEM decreasing trend in this region (China Sea, Japan, South Korea) as well? Cluster EAST explains 35% of the decline, i.e., 0.35 x 0.52 = 0.182 ng/m3/yr. Is that consistent with trends reported in this region (e.g., Kim et al., 2016)?

Lines 321-323: "We also noted that the largest decline of Hg concentrations was observed in the cluster SW, which indicated more effective air pollution control in the regions where the air mass of the cluster SW passed". What about the seasonality of GEM concentrations in the various clusters (NW, SW, EAST)? Could a difference in seasonality explain the observed Hg decline?

Figure 3: Could you please add the standard deviations? Is that the average over several years?

References

Chen, L., Zhang, W., Zhang, Y., Tong, Y., Liu, M., Wang, H., Xie, H., Wang, X., 2018. Historical and future trends in global source-receptor relationships of mercury. Sci. Total Environ. 610, 24-31. https://doi.org/10.1016/j.scitotenv.2017.07.182

Corbitt, E.S., Jacob, D.J., Holmes, C.D., Streets, D.G., Sunderland, E.M., 2011. Global Source-Receptor Relationships for Mercury Deposition Under Present-Day and 2050 Emissions Scenarios. Environ. Sci. Technol. 45, 10477-10484. https://doi.org/10.1021/es202496y

Kim, K.-H., Brown, R.J.C., Kwon, E., Kim, I.-S., Sohn, J.-R., 2016. Atmospheric mercury at an urban station in Korea across three decades. Atmos. Environ. 131, 124-132. https://doi.org/10.1016/j.atmosenv.2016.01.051

Martin, L.G., Labuschagne, C., Brunke, E.-G., Weigelt, A., Ebinghaus, R., Slemr, F., 2017. Trend of atmospheric mercury concentrations at Cape Point for 1995-2004 and since 2007. Atmos Chem Phys 17, 2393-2399. https://doi.org/10.5194/acp-17-2393-2017

Pacyna, J.M., Travnikov, O., De Simone, F., Hedgecock, I.M., Sundseth, K., Pacyna, E.G., Steenhuisen, F., Pirrone, N., Munthe, J., Kindbom, K., 2016. Current and future levels of mercury atmospheric pollution on a global scale. Atmos Chem Phys 16, 12495-12511. https://doi.org/10.5194/acp-16-12495-2016

Slemr, F., Angot, H., Dommergue, A., Magand, O., Barret, M., Weigelt, A., Ebinghaus, R., Brunke, E.-G., Pfaffhuber, K.A., Edwards, G., Howard, D., Powell, J., Keywood, M., Wang, F., 2015. Comparison of mercury concentrations measured at several sites in the Southern Hemisphere. Atmos Chem Phys 15, 3125-3133. https://doi.org/10.5194/acp-15-3125-2015

Soerensen, A.L., Skov, H., Jacob, D.J., Soerensen, B.T., Johnson, M., 2010. Global concentrations of gaseous elemental mercury and reactive gaseous mercury in the marine boundary layer. Environ. Sci. Technol. 44, 7425-7430.

Sprovieri, F., Pirrone, N., Bencardino, M., D'Amore, F., Carbone, F., Cinnirella, S., Mannarino, V., Landis, M., Ebinghaus, R., Weigelt, A., Brunke, E.-G., Labuschagne, C., Martin, L., Munthe, J., Wängberg, I., Artaxo, P., Morais, F., Barbosa, H.D.M.J., Brito, J., Cairns, W., Barbante, C., Diéguez, M.D.C., Garcia, P.E., Dommergue, A., Angot, H., Magand, O., Skov, H., Horvat, M., Kotnik, J., Read, K.A., Neves, L.M., Gawlik, B.M., Sena, F., Mashyanov, N., Obolkin, V., Wip, D., Feng, X.B., Zhang, H., Fu, X., Ramachandran, R., Cossa, D., Knoery, J., Marusczak, N., Nerentorp, M., Norstrom, C., 2016. Atmospheric mercury concentrations observed at ground-based monitoring sites globally distributed in the framework of the GMOS network. Atmos Chem Phys 16, 11915-11935. https://doi.org/10.5194/acp-16-11915-2016

Sung, J.-H., Roy, D., Oh, J.-S., Back, S.-K., Jang, H.-N., Kim, S.-H., Seo, Y.-C., Kim,
J.-H., Lee, C.B., Han, Y.-J., 2018. Trans-boundary movement of mercury in the Northeast Asian region predicted by CAMQ-Hg from anthropogenic emissions distribution. Atmospheric Res. 203, 197-206. https://doi.org/10.1016/j.atmosres.2017.12.015

Wu, Q., Wang, S., Li, G., Liang, S., Lin, C.-J., Wang, Y., Cai, S., Liu, K., Hao, J., 2016. Temporal Trend and Spatial Distribution of Speciated Atmospheric Mercury Emissions in China During 1978-2014. Environ. Sci. Technol. 50, 13428-13435. https://doi.org/10.1021/acs.est.6b04308

---

## Author Comment (AC1) · 9 May 2018

**Reply to Comments from Reviewer #1**

We thank the editor and reviewers' comments which help us to improve the manuscript. We have carefully revised our manuscript following the reviewers' comments. A point-to-point response is given below. The reviewers' comments are in black and our replies are in blue.

**To reviewer**

*Comment 1:*

The authors mention that mercury species were measured but only GEM data are presented and their trend calculated. What was the contribution of GOM and PBM? Could they contribute to the trend? This is important to discuss because the regional atmospheric Hg emissions in Section 2.5 are probably not only those of GEM but of total mercury. What was the seasonal variation of GOM and PBM? Could it provide some additional evidence for the reasons of GEM seasonal variation?

*Response:*

The average GOM and PBM concentrations during the studied period were 14.81 $\pm$ 13.21 pg m$^{-3}$ and 20.10 $\pm$ 34.02 pg m$^{-3}$, which accounted for 0.68% and 0.92% in total Hg, respectively. Therefore, the contribution of GOM and PBM to total Hg trend was supposed to be negligible. The downward trend of atmospheric Hg was dominated by the GEM concentration.

It is true that anthropogenic Hg emission inventories included GEM, GOM and PBM emissions. However, the residence time of GOM and PBM is shorter than that of GEM, generally several days to a few weeks for GOM and PBM and 0.5 – 2 year for GEM (Schroeder and Munthe, 1998). In addition, the concentrations of GOM and PBM were affected by emissions, weather condition and depositon processes simultaneously. Therefore, the GEM concentrations in the air of a background site are primarily impacted by GEM emissions. Thus, the regional atmospheric Hg emissions in Section 2.5 are GEM emissions instead of total Hg emissions. We have added sentences to make this point clear in Section 2.5.

Figure R1 showed the seasonal variation of GOM and PBM from March 2014 to February 2015. Considering that the concentrations of GOM and PBM were affected by emissions, weather condition and depositon processes simultaneously, we need more researches to determine the dominant impact factors. So it is hard to get some additonal evidence for the reason of GEM seasonal variation from the seasonal variation of GOM and PBM.

[Figure]

**Figure R1**. Monthly variation of GEM, GOM and PBM at Chongming from March 2014 to

February 2015

The calculation process of GEM emissions is revised as below.

"Regional atmospheric GEM emissions by month are calculated by using both the technology-based emission factor methods and transformed normal distribution function method. Detailed introduction of these two methods and the speciation profile of the emitted

Hg for each sector are described in our previous study (Wu et al., 2016)."

**See the revised manuscript, line 206-209**

*Comment 2:*

Section 3.1: Averages and their standard deviations should always be stated with the number of measurements because only then statistical tests for significance of differences can be made.

In line 194 the authors claim that GEM concentrations in 2014 were significantly higher than…

- at which significance level? Line 199: the annual decrease rate should be given with its uncertainty and number of months.

*Response:*

Both the standard deviation and the number of measurements have been added in the text for statistical tests of significance of differences.

The GEM concentrations in 2014 were significantly higher than the background concentration of Northern Hemisphere at the significance level with p value less than 0.01.

"The GEM concentrations in 2014 were higher (*t* test, $p<0.01$) than the Northern Hemisphere back-ground concentration (about 1.5 ng m$^{-3}$) (Sprovieri et al., 2010) and those measured in other remote and rural locations in China (Zhang H et al., 2015; Fu et al., 2008a; Fu et al.,

2009)."

**See revised manuscript, line 231 - 234.**

The annual decrease rate has been given with its uncertainty and the number of months. In addition, the number of valid data to calculate the monthly average is listed in Table S3.

"During this period, monthly GEM concentrations showed a significant decrease with a rate of

$-0.60 \pm 0.08$ ng m$^{-3}$ yr$^{-1}$ ($R^2$=0.64, $p<0.01$ significance level, n = 32) (Figure 2a)."

**See revised manuscript, line 236 - 237.**

          **Table S3.** The number of valid data during sampling period

**See revised manuscript, supporting information Table S3.**

***Comment 3:***

| Year | Jan | Feb | Mar | Apr | May | Jun | Jul | Aug | Sep | Oct | Nov | Dec |
|------|-----|-----|-----|-----|-----|-----|-----|-----|-----|-----|-----|-----|
| 2014 | | | 5914 | 6125 | 6493 | 5568 | 4634 | 6255 | 6491 | 7106 | 7578 | 5564 |
| 2015 | 5227 | 4532 | 5216 | 3392 | 4072 | 4797 | 7591 | 6538 | 3434 | 2223 | 4363 | 8833 |
| 2016 | | | 1370 | 8293 | 7476 | 5884 | 5424 | 5641 | 3561 | 4544 | 6292 | 4589 |

Figure 3: It is not clear how the points in Figure 3 were calculated? Were the data detrended before the averaging? In view of the strong downward trend they should be. What is the standard deviation or standard error of the monthly means – please show them as vertical bars in Fig. 3. Are the differences between the months statistically significant? This is a precondition for the discussion of the seasonal variation.

***Response:***

The GEM concentrations in the same month but different years were averaged to get monthly average during sampling period. The monthly GEM concentrations were detrended before the average (Figure 2). The standard deviations of the monthly means have been added.

"According to the decomposition result (Figure 2c), we observed strong seasonal cycle with seasonal GEM peak in July and trough in September, so GEM concentrations in the same month but different years were averaged to discuss the seasonal circle (Figure 3). The average data can eliminate the effect of downward trend and get result of average seasonal variation. The error bars in the Figure 3 mean the standard deviation of the monthly average."

**See revised manuscript, line 274 –278.**

[Figure]

**Figure 3.** Monthly variations of GEM concentration at remote sites in China

**See revised manuscript at Figure 3.**

The difference between month are statistically significant (F test, p<0.001). In addition, the observed GEM concentration signal was decomposed (signal = trend + seasonal + random, example here: https://anomaly.io/seasonal-trend-decomposition-in-r/). By using this method, we also observed very strong detrended seasonal cycle where GEM peak was observed in July and the GEM trough was in September.

[Figure]

**Figure 2.** Monthly average GEM concentrations during the studied period (a) observed monthly

GEM concentrations (b) GEM trend after decomposition (c) GEM seasonality after decomposition (d) GEM random after decomposition

Note: The observed concentrations during July 2015-April 2016 were TGM concentrations indeed due to the problems of Tekran 1130/1135. However, the GOM concentrations at Chongming island accounted for less than 1% of TGM. Thus, the GEM concentrations were approximated to TGM

concentrations during July 2015-April 2016.

***Comment 4:***

4.1 Figure 4 and its capture: This figure needs substantial revision to illustrate the point the authors make and to make it understandable for the readers. Negative emissions are deposition fluxes and should be named as such. Thus "natural emissions" in spring, autumn and winter are in fact deposition fluxes. Net fluxes are needed to illustrate the point made by the authors but they are not shown. The capture should also state that it is about the emissions and depositions in the YRD region?

***Response:***

The natural emissions in the manuscript are defined as the followings.

Nature emissions=bi-directional Hg flux × studied period × studied area.

Therefore, emissions and flux are different concepts in this study. We use "natural emissions"

instead of "natural flux" to correspond to "anthropogenic emissions". It should be pointed out that the natural emission is a concept of net natural emission, which reflected a net effect of two competing processes (Zhang, 2009): total natural Hg emissions and total Hg deposition.

When the value is positive, it means the net effect is Hg emissions to air. Otherwise, Hg deposites. We have made this concept clear in both text and Figure 4.

"The GEM emissions from natural sources $E_N$ are calculated as followings.

$$E_N = \sum_i F_i \times A_i \times t \tag{6}$$

Where $F_i$ is a bi-directional Hg flux of canopy $i$, ng km$^{-2}$ yr$^{-1}$; $A$ is the studied area, km$^{-2}$; $t$ is the studied year, yr. The bi-directional Hg flux was obtained from the study of Wang et al. (2016)

directly. It should be pointed out that the natural emission is a concept of net emission in this manuscript, which reflected a net effect of two competing processes (Zhang, 2009): total Hg natural emissions and total Hg deposition. The total natural emissions included primary natural release and re-emission of legacy Hg stored in the terrestrial and water surface (Wang et al.,

2016). When the value is positive, it means the net effect is Hg emissions to air. Otherwise, Hg deposited."

**See revised manuscript at line 219 – line 227.**

[Figure]

**Figure 4.** Seasonal cycle of GEM concentrations and natural emissions during 2014-2016. The error bars represent the standard deviation of seasonal average. Positive values of natural emissions represent Hg emitted to air. Otherwise, Hg deposited.

**See the revised manuscript Figure 4.**

4.2 This will rise a problem: trajectory analysis in section 3.4 shows large influence of transport from the NW provinces of China outside of the YRD region. How does this transport influence the seasonal variation?

*Response:*

Trajectory outside of the YRD region showed similar seasonal variation as those passes through

YRD region. The original definition of "NW" "SW" and "EAST" did not clearly distinguish the pathway of trajectory. Thus, the original "NW" actually contains the Jiangsu province of

YRD region. In our revised manuscript, the trajectories were grouped into 3 clusters: NCP, SW-

YRD and ABROAD. The NCP mainly passed through north China plain and around regions but not via the YRD regions; the SW-YRD passed through the YRD regions before arriving to

Chongming island; the ABROAD mainly originated from the East China Seas, South Korea,

Japan and Northeast Asia continent, and then arrived to our monitoring sites directly without passing the mainland China. From Figure S3, we can see that the NCP cluster and

ABROAD cluster showed similar seasonal variation as cluster SW-YRD. High GEM

concentrations were observed in summer.

[Figure]

**Figure S3.** The seasonality of GEM concentration in the NCP, SW-YRD and ABORD region (No trajectory transport though ABROAD in winter of 2014)

We also revised the manuscript as below.

"Transport also overall enhanced the observed seasonal variation of GEM concentrations at

Chongming Island. According to the statistics of backward trajectories in section 3.4, the GEM

concentrations in the air mass which did not pass via the YRD regions also showed high GEM

concentration in warm season in 2014 (Figure S3)."

**See the revised manuscript at line 314 - 317**

*Comment 5:*

Line 243-245: "The annual emissions from both natural source and anthropogenic source…

was 0.75 and 10.3 t, respectively" – the reader may think that natural emissions make less than

10% of the anthropogenic ones and cannot thus be responsible for the seasonal variation. One has to look in Fig. 4 to find out that the "natural emissions" are a sum of natural emissions in one season and "natural" deposition fluxes in three seasons. That provokes a question: how is anthropogenically emitted mercury removed from the atmosphere if there are only "natural"

deposition fluxes? Please use the correct terminology and separate the natural and anthropogenic emissions from the deposition fluxes of both.

***Response:***

Sorry for the misunderstanding. To avoid confusion, we deleted the sentence of "The annual emissions from both natural source and anthropogenic source… was 0.75 and 10.3 t, respectively" in the revised manuscript. The impact of anthropogenic emissions and natural emissions were discussed separately. The anthropogenic emissions were in the range of 2.5 –

2.7 t while natural emissions varied from -5.4 – 8.4 t in different season. Thus, one important conclusion of our study is that the seasonal GEM cycle was dominated by the natural emissions.

"Source emission is one significant factor on GEM concentrations in the air. The GEM

concentrations at a remote site are generally regarded under the impact of regional emissions.

Therefore, the emissions in the YRD regions (Anhui, Zhejiang, Jiangsu, and Shanghai) were calculated. However, the anthropogenic emissions were in the range of 2.5-2.7 t, which is almost unchanged. Compared to the anthropogenic emissions, we observed almost synchronized trends between natural emissions and air Hg concentrations in Figure 4."

**See the revised manuscript at line 296 - 301**

It is difficult to distinguish whether the deposited Hg is from natural sources or anthropogenic sources. Therefore, the bi-directional Hg flux scheme contained total Hg deposition flux (both so called "natural" and "anthropogenic" deposition fluxes) (Zhang et al., 2009). And the natural

Hg emissions in this study have considered the removal of anthropogenic Hg emissions. We have clearly defined the concept of "natural emissions" in this study. The natural emission is a concept of net natural emission, which reflected a net effect of total natural Hg emissions and total Hg deposition amount. Therefore, the data of natural emissions in the four seasons contains both emissions and deposition. The positive value in summer means that net effect is Hg emissions to air. In other three seasons, Hg deposited. Detailed revision is described as follows.

"The GEM emissions from natural sources $E_N$ are calculated as followings.

$$E_N = \sum_i F_i \times A_i \times t$$

where $F_i$ is a bi-directional Hg flux of canopy $i$, ng km$^{-2}$ yr$^{-1}$; $A$ is the studied area, km$^{-2}$; $t$ is the studied year, yr. The bi-directional Hg flux was obtained from the study of Wang et al. (2016)

directly. It should be pointed out that the natural emission is a concept of net emission in this manuscript, which reflected a net effect of two competing processes (Zhang, 2009): total Hg natural emissions and total Hg deposition. The total natural emissions included primary natural release and re-emission of legacy Hg stored in the terrestrial and water surface (Wang et al.,

2016). When the value of $E_N$ is positive, it means the net effect is Hg emissions to air. Otherwise,

Hg deposited."

**See the revised manuscript at line 219 - 227**

***Comment 6:***

The results of PCA analysis and Table 2: The authors attribute the factor 2 to "exchange of

PBL with free troposphere" but do not explain why. Last row in the table 2 called "variance explain" lists for 2016 exchange of PBL with the free troposphere 75.735 which together with

50.625 for "combustion" makes more than 100. As such the units of "variance explain" cannot be percent. What are the numbers in this row and does it make sense to present them with three valid numbers after decimal point?

***Response:***

We have added explanation as follows.

"Considering the low loading of CO and high loading of $O_3$, the factor 2 can be viewed as a sign of the invasion of air mass from stratosphere (Fishman and Seiler, 1983; Jaffe, 2010). The air mass from stratosphere will increase the $O_3$ concentration. $O_3$ react with NO, which makes a negative correlation with NO. However, the low loading on GEM of factor 2 indicated that

Factor 2 had no relationship with GEM concentrations at Chongming from the aspect of whole year data."

**See the revised manuscript at line 350 - 355**

Sorry for the typo. We have corrected this in Table 1. The variance explain showed the contribution ratio of factor 1 and factor 2 in the total variance. We have revised the data to two valid numbers after decimal point.

**Table 1**. PCA component loading of GEM and other air pollutants

| Air pollutants | 2014 | | Air pollutants | 2016 | |
|---|---|---|---|---|---|
| | Factor 1 | Factor 2 | | Factor 1 | Factor 2 |
| $SO_2$ | **0.76** | 0.14 | $SO_2$ | **0.82** | -0.09 |
| $NO_X$ | **0.76** | -0.20 | $NO_X$ | **0.70** | **-0.52** |
| $O_3$ | -0.11 | **0.98** | $O_3$ | -0.41 | **0.97** |
| $PM_{2.5}$ | **0.85** | 0.05 | $PM_{2.5}$ | **0.88** | 0.05 |
| GEM | **0.66** | 0.02 | GEM | **0.78** | -0.19 |
| CO | **0.79** | 0.12 | | | |
| Component | Combustion | Invasion of air mass from stratosphere | Component | Combustion | Invasion of air mass from stratosphere |
| Variance explain (%) | 49.36 | 17.53 | Variance explain (%) | 50.63 | 25.10 |

Note: Text in bold phase were regarded as high loading (factor loading>0.40 or <-0.40)

***Comment 7:***

Table 3: The numbers are probably annual emissions but the capture does not say it. The year of the emissions is not given. I wonder about the "other SO2 sources" which are substantially larger than all coal, oil, and biomass burning taken together. If it is not an error, what are the

"other SO2 sources"?

***Response:***

Thank for the comments. It is the emission in 2014. The emission for "other $SO_2$ sources" is a typo and we have revised this. The other sectors contain municipal solid incineration, copper smelting, aluminum production, gold production, other coal combustion, stationary oil combustion, and cremation. Table 2 has been revised as follows.

**Table 2.** Emissions of the main air pollutants in YRD region in 2014

| Sectors | Annual emissions | | | |
|---|---|---|---|---|
| | $SO_2$ (kt) | $NO_x$ (kt) | $PM_{2.5}$ (kt) | GEM (t) |
| Coal-fired power plants | 918.31 | 991.62 | 118.42 | 14.00 |
| Coal-fired industrial boilers | 311.03 | 271.94 | 79.91 | 9.80 |
| Residential coal combustion | 68.48 | 42.11 | 163.93 | 0.40 |
| Cement clinker production | 207.48 | 371.13 | 208.02 | 4.70 |
| Iron and steel production | 480.97 | 142.80 | 169.84 | 2.30 |
| Mobile oil combustion | 38.43 | 1786.74 | 98.00 | 1.90 |

| Other sectors | 348.83 | 316.28 | 382.48 | 2.50 |

"The studied emission sectors included coal-fired power plants, coal-fired industrial boilers, residential coal-combustion, cement clinker production, iron and steel production, and other small emission sectors (eg., zinc smelting, lead smelting, municipal solid incineration, copper smelting, aluminum production, gold production, other coal combustion, stationary oil combustion, and cremation)."

**See the revised manuscript at line 213 - 216**

*Comment 8:*

Chapter 3.4 misses a major point: Table 3 of $SO_2$, $NO_x$, $PM_{2.5}$, and GEM emissions is only for one undefined year and only for the YRD region. Table 4 and Figure 6 show a dominant influence of transport from NW of China which is mostly outside of the YRD region. To illustrate convincingly the major conclusion of the paper one would need a table with the inventories for NW and SW (perhaps separately) and for 2014 and 2016.

*Response:*

According to our response to comment 4, we have adjusted the cluster. Based on the adjusted cluster, we added the emissions of $SO_2$, $NO_x$ and $PM_{2.5}$ in both 2014 and 2016 to illustrate the change of emission inventory in NCP and SW-YRD region (Table S5). According to the table, we observed obvious emission decline of the above pollutants.

**Table S5.** Emission inventories of the main pollutants from the studied regions in 2014 and 2016

| Air pollutants | 2014 | | 2016 | | Decline proportion | |
|---|---|---|---|---|---|---|
| | NCP | SW-YRD | NCP | SW-YRD | NCP | SW-YRD |
| $PM_{2.5}$ (kt) | 2019 | 1209 | 1849 | 1109 | -8% | -8% |
| $NO_x$ (kt) | 5697 | 4022 | 5424 | 3855 | -5% | -4% |
| $SO_2$ (kt) | 3780 | 1993 | 3450 | 1780 | -9% | -11% |
| GEM (t) | 118 | 72 | 103 | 67 | -13% | -7% |

Note: According to the contribution of trajectory, the dominant provinces in the NCP region included Beijing, Tianjin, Hebei, Shandong and Liaoning province. The SW-YRD mainly contained Shanghai, Zhejiang, Jiangsu, Jiangxi and Anhui province.

**See the revised manuscript at line 382 - 384**

*Comment 9:*

Table 1: The paper is about regional trend and I wonder why it is necessary to discuss global background trends in such detail. Also because the reasons of downward trend of mercury at many background stations of the world at the time of increasing global emissions is still not well understood (compare Horowitz et al., EST 48, 10242-10250, 2014, with Soerensen et al.,

GRL, 39, L21810, doi:10.1029/2012GL053736, 2012).

*Response:*

We agree with your valuable comment. Our paper is about regional trend, so we have curtailed the discussion and move the original Table 1 to supporting information (Table S4) so as to focus on our topic.

*Comment 10:*

Figure S1: Here the least square fit of 4 points provides R2 of 0.487 for January and 0.613 for

February for which the authors claim $p < 0.01$ in both cases. This is surely incorrect because 4

points result statistically in only 2 degrees of freedom. Please explain.

*Response:*

The purpose of original Figure S1 is to obtain the fitting curve so as to calculate the Hg concentrations in the January and February of 2016. But it did not work actually because the points are not enough. Therefore, we deleted the original Figure S1. Instead, the average value of the GEM concentrations in January of 2015 and 2017 were used to represent the GEM

observation in January of 2016. The same method is used to simulate the GEM observation in

February of 2016.

The Figure S1 was revised as below.

[Figure]

**Figure S1.** The trend of monthly average GEM concentration from March 2014 to February

2017. The monthly average of GEM in January of 2016 is simulated as the average value that in the January of 2015 and 2017. The same method is used for the data in February of 2016.

*Comment 11:*

Figure S2: The downward annual rate should be given with its standard error.

*Response:*

The downward annual rate was given with its standard error as follows. See the Figure S1 in the response of comment 10.

*Comment 12:*

Line 51: "Both GOM and PBM are more soluble." than what? PBM is not necessarily more soluble than GEM but it is scavenged by wet deposition. Low solubility of GEM need to be mentioned before this statement.

*Response :*

The statement has been revised as follows.

"GOM is much soluble than GEM, and PBM can be quickly scavenged by both dry and wet deposition. Therefore, the residence time of both GOM and PBM is shorter than that of GEM, generally several days to a few weeks for GOM and 0.5 – 2 year for GEM (Schroeder and

Munthe, 1998). "

**See the revised manuscript at line 52 - 55.**

*Comment 13:*

Lines 71/72: …have been estimated to decrease…

*Response:*

We have revised the manuscript as follow.

"However, recently atmospheric Hg emissions in China have been estimated to decrease since

2012 (Wu et al., 2016)."

**See the revised manuscript at line 82**

*Comment 14:*

Line 86: …is located…

*Response:*

We have revised the manuscript as suggested.

"As China's third largest island, Chongming Island is located in the east of Yangtze River Delta region with a typical subtropical monsoon climate."

**See the revised manuscript at line 97**

*Comment 15:*

Lines 102/103: … the error between gold trap A and gold trap B was limited to…? Probably the difference instead of error was limited. What happens if the difference is more than the limit?

*Response:*

We have replaced the error with differences. If the difference is more than the limit, it means that the gold traps are passivizing and we need to replace the old gold trap A and gold trap B.

*Comment 16:*

Lines 171/172: "uncertainties" would be better than "errors"

*Response:*

We have revised the manuscript as suggested.

*Comment 17:*

Line 199: Please state the decrease rate with its standard error.

*Response*

We have revised the manuscript as suggested.

*Comment 18:*

Lines 207-209: A reference to Martin et al (2017) is not correct because the paper does not contain annual averages and the authors of this paper do not mention a gap in the measurements between 2004 and 2007. The correct reference would be: annual average GEM concentration decreased from 1.29 ng m-3 in 1996 to 1.19 ng m-3 in 2004 (Slemr et al., GRL 35, L11807, doi:10.1029/2008GL033741, 2008) and were increasing from 0.93 ng m-3 in 2007 (Slemr et al., ACP 15, 3125-3133, 2015) until 2016 (Martin et al, 2017).

*Response:*

We have revised the manuscript as suggested.

"In South Africa, annual average GEM concentration at Cape Point decreased from 1.29 ng

$m^{-3}$ in 1996 to 1.19 ng $m^{-3}$ in 2004 (Slemr et al., 2008) and were increasing from 0.93 ng $m^{-3}$

in 2007 (Slemr et al., 2015) until 2016 (Martin et al, 2017)."

**See the manuscript at line 261– 263**

**Figure 2.** Monthly average GEM concentrations during the studied period (a) observed monthly

GEM concentrations (b) GEM trend after decomposition (c) GEM seasonality after decomposition (d) GEM random after decomposition

Note: The observed concentrations during July 2015-April 2016 were TGM concentrations indeed due to the problems of Tekran 1130/1135. However, the GOM concentrations at

Chongming island accounted for less than 1% of TGM. Thus, the GEM concentrations were approximated to TGM concentrations during July 2015-April 2016.

The revised manuscript of decomposition trend is revised as below.

"From another aspect, the trend decomposition of the GEM concentration signal (signal =

trend + seasonal + random) from March 2014 to December 2016 were performed in Figure 2

(https://anomaly.io/seasonal-trend-decomposition-in-r/). By using this method, we also observed a pronounced trend (Figure 2b) and the random was limited in the range of -0.24 –

0.24 ng m$^{-3}$ (Figure 2d)."

**See the revised manuscript at line 238 - 243**

"From Figure 1, we also observed more pronounced seasonal variation in 2014, which can be attributed to the lower wet deposition and GEM oxidation. On one aspect, as a costal site, the Chongming Island is abundant with •OH. The increase of $O_3$ concentration from the summer of 2014 to 2016 may contribute to a higher oxidation of GEM in 2016. On another aspect, and higher wet Hg deposition is approximately 6.6 times of that in the winter at Chongming (Zhang et al., 2010). Meanwhile, the rainfall in 2016 summer (546 mm) was higher than the rainfall in

2014 (426 mm). Therefore, the higher oxidation and wet deposition rate of Hg in the summer of 2016 will reduce the concentration difference between summer and winter, which lead to a less pronounced seasonal variation in 2016. Meanwhile, the higher oxidation and wet deposition in 2016 also contributed to the downward trend of GEM by reducing the seasonality in spring and summer (Figure S3).”

**See the revised manuscript at line 318 - 328**

While $SO_2$, $NO_2$, and PM concentrations were monitored, data are not presented nor discussed. Do you also observe a decreasing trend? That would be the best way to support that

"air pollution control policies targeting $SO_2$, $NO_2$, and PM reductions had significant co- benefits on atmospheric Hg".

*Response:*

To further verify the cause of downward trend of atmospheric Hg, we give the emission inventory (Table S6) and concentrations (Table S5) of other air pollutants in the studied regions in both 2014 and 2016. Both the emissions and concentrations of $SO_2$, $NO_2$, and PM showed a decreasing trend, which is used to support that "air pollution control policies targeting $SO_2$,

$NO_2$, and PM reductions had significant co-benefits on atmospheric Hg".

“Table 3 showed the detailed data of the three classifications. From 2014 to 2016, the whole

China region (NCP, SW-YRD) contributed to 70% of GEM decline at Chongming Island.

Considering downward trend of emission inventory and atmospheric pollutant from 2014 to

2016 in NCP and SW-YRD region (Table S5, Table S6), the reason of downward trend can be attributed to the effectiveness of existing air pollution control measures in China (SC, 2013;

MEP, 2014).”

**See the revised manuscript at line 380 – 384**

**Table S5.** The annual concentration of $SO_2$, $NO_x$, $O_3$ and $PM_{2.5}$ at Chongming site, NCP, and

SW-YRD regions

| Year | | 2014 | | | 2016 | | | Change | | |
|------|--------|------|--------|-----------|------|--------|-----------|------|--------|-----------|
| Pollutants | Region | NCP | SW-YRD | Chongming | NCP | SW-YRD | Chongming | NCP | SW-YRD | Chongming |
| PM$_{2.5}$ (μg m$^{-3}$) | | 71.93 | 53.05 | 25.09 | 60.75 | 44.75 | 23.89 | -16% | -16% | -5% |
| SO$_2$ (μg m$^{-3}$) | | 34.52 | 21.01 | 1.60 | 24.37 | 16.40 | 1.47 | -29% | -22% | -8% |
| NO$_2$ (μg m$^{-3}$) | | 45.07 | 34.34 | 12.62 | 41.55 | 34.40 | 10.84 | -8% | 0% | -14% |
| O$_3$ (μg m$^{-3}$) | | 60.29 | 56.27 | 41.70 | 61.84 | 60.92 | 44.38 | 3% | 8% | 6% |
| GEM (ng m$^{-3}$) | | No data | | 2.68 | No data | | 1.60 | No data | | -40% |

Note: According to the contribution of trajectory, the dominant provinces in the NCP region
included Beijing, Tianjin, Hebei, Shandong and Liaoning province. The SW-YRD mainly
contained Shanghai, Zhejiang, Jiangsu, Jiangxi and Anhui province.

**Table S6.** Emission inventories of the main pollutants from the studied regions in 2014 and
2016

| Air pollutants | 2014 | | 2016 | | Decline proportion | |
|----------------|------|--------|------|--------|------|--------|
| | NCP | SW-YRD | NCP | SW-YRD | NCP | SW-YRD |
| PM$_{2.5}$ (kt) | 2019 | 1209 | 1849 | 1109 | -8% | -8% |
| NO$_x$ (kt) | 5697 | 4022 | 5424 | 3855 | -5% | -4% |
| SO$_2$ (kt) | 3780 | 1993 | 3450 | 1780 | -9% | -11% |
| GEM (t) | 118 | 72 | 103 | 67 | -13% | -7% |

Note: According to the contribution of trajectory, the dominant provinces in the NCP region
included Beijing, Tianjin, Hebei, Shandong and Liaoning province. The SW-YRD mainly
contained Shanghai, Zhejiang, Jiangsu, Jiangxi and Anhui province.

Finally, I wonder why GOM and PBM data are not reported and discussed. Do you also observe a decreasing trend? You may have encountered issues with the speciation unit. If so, was the experimental setup identical in 2014 and 2016, or did you analyze GEM when the speciation unit was working vs. TGM when it wasn't? A discussion on analytical uncertainties would be much welcomed.

The GOM and PBM data are not reported finally due to the following reasons. First, the main purpose of this manuscript is to validate the anthropogenic Hg emissions reduction through observation data. The Chongming sit is a background site. Considering the stability of GEM in the air, we choose GEM as an index to reflect the emission control effect. Second, the method we developed to explain the GEM trend is not applicable for GOM and PBM. Except emissions, we think the potential reactions in the air are significant factors to impact both GOM and PBM

concentrations. But we need more evidence to prove our assumptions. Therefore, we deleted the discussion of GOM and PBM in our final manuscript.

We also observed decreasing trend of PBM. But the GOM kept increasing. Currently, we need more study to explain this phenomenon.

Yes, we also encountered issues with the speciation unit. From July 5, 2015 to April 30 2016, the Tekran 1130/1135 speciation unit was damaged by the rainstorm, the Tekran 2537X were operated without speciation units but with PTFE filter to protect the instrument from particles and sea salt. The average concentration of GOM and PBM during sampling period was 14.81

$\pm 13.21$ and $20.10 \pm 34.02$ pg m$^{-3}$, respectively. The GOM fraction in TGM was less than 1%

at Chongming island (Li et al., 2016; Zhang et al., 2017). Therefore, the GEM concentrations were approximated to TGM concentrations July 5, 2015 to April 30 2016 when the speciation unit does not work, as most of other studies have done (Slemr et al., 2015; Sprovieri et al.,

2016). In addition, uncertainty analysis was performed to point out potential impact. The revision is as follows.

"From July 5, 2015 to April 30 2016, the Tekran 1130/1135 speciation unit was damaged by the rainstorm, the Tekran 2537X were operated without speciation units but with PTFE filter to protect the instrument from particles and sea salt. Therefore, the observed concentrations during

July 2015-April 2016 were TGM concentrations indeed. However, the GOM concentrations at

Chongming Island accounted for less than 1% of TGM (TGM=GOM+GEM). Thus, the GEM

concentrations were approximated to TGM concentrations from July 2015 to April 2016."

**See the revised manuscript at line 109 – 115**

"In our research, random uncertainties of individual measurement had been averaged out and the systematic uncertainties need to be considered. The overall practically achievable systematic uncertainty would be 10% considering that the instrument was not in ideal performance (Slemr et al., 2015; Steffen et al., 2012). For example, slow deactivation of the traps, contamination of the switching valves and leaks would increase the uncertainties but were difficult to quantify (Slemr et al., 2015;Steffen et al., 2012). Because of the consistency of instrument and the quality assurance/quality control have been paid special attention to during the sampling campaign, the systematic differences of instrument did not affect the huge variation between 2014 and 2016."

**See the revised manuscript at line 124 – 131**

*Comment 1:*

Line 26: "GEM concentrations showed a significant decrease with a rate of -0.60 ng/m3/yr".

According to Table 1, the rate is -0.52 ng/m3/yr.

*Response:*

Thanks for the comments. This is a typo in the table. It is $-0.60 \pm 0.08$ ng m$^{-3}$ yr$^{-1}$ actually. We have corrected this in the revised manuscript.

.

*Comment 2:*

Line 33: "It was find" should be "It was found".

*Response:*

We have corrected in the manuscript as below.

"It was found that the reduction of domestic emissions was the main driver of GEM decline in Chongming Island, accounting for 70% of the total decline."

**See the revised manuscript at line 33 – 34**

*Comment 3:*

Lines 47-48: "In the atmosphere, Hg mainly presents as GEM, accounting for over 95% or the total". Can you please add a reference? Is that also true at your site?

*Response:*

We have added references in the manuscript. During the sampling period (March 2014 to

June 2015 and May 2016 to December 2016), the GOM concentration is $14.81 \pm 13.21$ pg m$^{-3}$

and GEM concentration is $2.15 \pm 0.94$ ng m$^{-3}$. Thus, the GOM concentration accounted for

0.68% and the conclusion in the reference is also true at our site. We also added reference in our manuscript as below.

"In the atmosphere, Hg mainly presents as GEM, accounting for over 95% of the total in most observation sites (Fu et al., 2015; Li et al., 2016; Zhang et al., 2017)."

**See the revised manuscript at line 49**

*Comment 4:*

Line 61-62: "(: : :) there is no official national monitoring network of atmospheric Hg". Out of curiosity, what is the current status of the Asian-Pacific Mercury Monitoring Network (http://nadp.sws.uiuc.edu/newIssues/asia/)? Do you think that Chinese sites will be included?

*Response:*

Thanks for the comment. Asian-Pacific Mercury Monitoring Network (APMMN) was established in 2013 with founding countries and regions including the U.S, China Taiwan,

Thailand,     Vietnam,     Indonesia,     Japan,     Korea     and     Canada (http://apmmn.org/AboutAPMMN.html). APMMN has been monitoring atmospheric mercury deposition in the Asia-Pacific region and holds annual meetings since 2013. Currently, there is no monitoring site of mainland China in the APMMN (see the Figure R2).

[Figure]

Figure R2. The participating country of APMMN (http://apmmn.org/AboutAPMMN.html)

Considering the large Hg emissions in mainland China, including the Chinese sites into the monitoring network will helpful for the research of Hg behavior in the regional or global scale.

However, nearly all mercury monitoring sites belong to individual researchers in China currently. Therefore, whether the Chinese sites will be included mainly depend on multiple factors such as individual interests and potential benefit. We also revised our expression in the manuscript as below.

"For the developing countries such as China, limited atmospheric Hg observations have been carried out (Fu et al., 2008b; Zhang H et al., 2016; Hong et al., 2016) and there is no official national monitoring network of atmospheric Hg in mainland China."

**See the revised manuscript at line 63 - 66**

*Comment 5:*

Lines 64-67: "Atmospheric Hg emissions in China accounted for 27% of the global total in

2010 (UNEP, 2013), which led to high air Hg concentrations in China. Therefore, atmospheric

Hg observations in China are critical to understand the Hg cycling at both regional and global scale". Please define "high" air Hg concentrations. Additionally, in order to emphasize the fact that observations in China are critical to understand the Hg cycling on a global scale, you could perhaps add a sentence about 1) future projections (e.g., Chen et al., 2018; Pacyna et al., 2016), and 2) long-range transport of Chinese emissions to other regions (e.g., Chen et al., 2018;

Corbitt et al., 2011; Sung et al., 2018).

*Response:*

Hg concentration in remote site in China and Northern Hemisphere are compared to illustrate the level of Hg pollution in China. Long-range transport and future projections are added to emphasize observation in China are critical to understand Hg cycling on a global scale.

The related paragraph is revised as below.

"China contributes to the largest Hg emissions in the world and will continue to be one significant Hg emitter for global Hg emissions in the coming future (UNEP, 2013, Wu et al.,

2016, Chen et al., 2018; Pacyna et al., 2016). Large Hg emissions in China have led to the average air Hg concentrations of $2.86 \pm 0.95$ ng m$^{-3}$ (in the range of 1.60-5.07 ng m$^{-3}$) at the remote sites in China (Fu et al., 2015). Such Hg concentration level is approximately 1.3 ng m$^-$

$^3$ higher than the background concentration of GEM in Northern Hemisphere (Zhang et al.,

2016;Sprovieri et al., 2017;Fu et al., 2015). In addition, the large Hg emissions in China will also impact the air Hg concentrations in East Asia and even North America through long-range transport (Sung et al., 2018;Zhang et al., 2017)."

**See the revised manuscript at line 67 – 75.**

*Comment 6:*

Lines 93-94: "we used Tekran 2537X/1130/1135 instruments to monitor speciated Hg in the atmosphere". I wonder why GOM and PBM concentrations are not reported in the manuscript.

*Response:*

We didn't report the GOM and PBM concentration because the main purpose of this manuscript is to validate the anthropogenic Hg emissions reduction through observation data.

The Chongming site is a background site. Considering the stability of GEM in the air, we choose GEM as an index to reflect the emission control effect. In addition, the method we developed to explain the GEM trend is not applicable for GOM and PBM. Except emissions, we think the potential reactions in the air are more significant factors for GOM and PBM. But we need more evidence to prove our assumptions. Therefore, we deleted the discussion of GOM

and PBM in our final manuscript.

If concentrations were recorded, it would be interesting to discuss the results. Do you also see a decreasing trend from 2014 to 2016? From 1978 to 2014, the fractions of GEM and PBM

decreased, while the GOM emission share gradually increased (Wu et al., 2016). What about the speciation of emissions since 2014? Can you observe a trend in GOM/PBM concentrations?

*Response:*

Yes. The downward trend of PBM concentration was observed to decrease from 2014 (24.51

$\pm$43.31 pg m$^{-3}$) to 2016 (22.07 $\pm$30.55 pg m$^{-3}$), which was also consistent with the downwawrd trend of GEM. However, the GOM concentration increased from (15.41 $\pm$ 16.02 pg m$^{-3}$) in

2014 to (18.97 $\pm$ 9.28 pg m$^{-3}$) in 2016. Speciated Hg emissions were showed in Table R1. All speciated Hg emissions have decreased since 2014 in the YRD regions. However, we observed significant GEM decreasing. But the decrease of GOM and PBM was quite slight.

**Table R1.** Speciated Hg emissions in YRD region and concentration at Chongming island in

2014 and 2016

| Year | Emission | | | Concentration | | |
|------|----------|----------|----------|----------|----------|----------|
| | GEM (t) | GOM (t) | PBM (t) | GEM (ng m$^{-3}$) | GOM (pg m$^{-3}$) | PBM (pg m$^{-3}$) |
| 2014 | 34.26 | 30.41 | 1.50 | 2.68 | 15.41 | 24.51 |
| 2016 | 27.65 | 29.16 | 1.39 | 1.60 | 18.97 | 22.07 |

Alternately, did you have issues with the speciation unit? It is quite common and I would appreciate an open discussion about that and associated analytical uncertainties. What kind of issues did you encounter? Are you confident that you collected and analyzed GEM (vs. TGM)

during the entire experiment? Was the instrumental setup exactly the same during the entire experiment? If not, how can you compare GEM concentrations without discussing analytical uncertainties? See major comment.

Yes, we encountered issues with the speciation unit. The Tekran 2537X was consistent and in good condition during the sampling period. There was no data in January and February in

2016 due to equipment failure. The Tekran 1130/1135 was accidentally rained by rain, so there was no data of speciated mercury between July 2015 and April 2016. From July 2015 to April

2016, we used Tekran 2537X only with PTFE filter to monitor atmospheric mercury. The average concentration of GOM during sampling period (March, 2014 to June 2015, May 2016

to December 2016) was 14.81 $\pm$ 13.21 pg m$^{-3}$, which is approximately 1% of GEM

concentration. Thus, the GEM concentrations were approximated to TGM concentrations July

5, 2015 to April 30 2016 when the speciation unit does not work, as most of other studies have done (Slemr et al., 2015; Sprovieri et al., 2016). In addition, we have added discussion about the analytical uncertainties to point out potential impact.

The manuscript was revised as below.

"From July 5, 2015 to April 30 2016, the Tekran 1130/1135 speciation unit was damaged by the rainstorm, the Tekran 2537X were operated without speciation units but with PTFE filter to protect the instrument from particles and sea salt. Therefore, the observed concentrations during

July 2015-April 2016 were TGM concentrations indeed. However, the GOM concentrations at

Chongming Island accounted for less than 1% of TGM (TGM=GOM+GEM). Thus, the GEM

concentrations were approximated to TGM concentrations from July 2015 to April 2016."

**See the revised manuscript at line 109 – 115**

"In our research, random uncertainties of individual measurement had been averaged out and the systematic uncertainties need to be considered. The overall practically achievable systematic uncertainty would be 10% considering that the instrument was not in ideal performance (Slemr et al., 2015; Steffen et al., 2012). For example, slow deactivation of the traps, contamination of the switching valves and leaks would increase the uncertainties but were difficult to quantify (Slemr et al., 2015;Steffen et al., 2012). Because of the consistency of instrument and the quality assurance/quality control have been paid special attention to during the sampling campaign, the systematic differences of instrument did not affect the huge variation between 2014 and 2016."

**See the revised manuscript at line 124 – 131**

*Comment 7:*

Lines 103-104: "The impactor plates and quartz filter were changed in every two weeks. The quartz filter was changed once a month". Did you change the quartz filter every two weeks or once a month?

*Response:*

Yes, the impactor plates, Teflon filter and quartz filter were changed in every two weeks.

The soda lime was changed once a month. We have corrected this sentence in the revised manuscript as below.

"The impactor plates and quartz filter were changed in every two weeks. The soda lime was changed once a month."

**See the revised manuscript at line 120 – 121**

*Comment 8:*

Line 106: "During the sampling campaigns, $PM_{2.5}$, $O_3$, $NO_x$, CO and $SO_2$ were monitored".

Why aren't you discussing the data, especially $SO_2$, $NO_x$, $PM_{2.5}$ while your main conclusion is that Hg decreasing trend in due to air pollution control policies targeting $SO_2$, $NO_x$, and $PM_{2.5}$.

I agree that you present emissions inventories, but I would really appreciate to see a real interpretation and discussion of these data. Do you also observe a decreasing trend? See major comment.

*Response:*

To further verify the cause of downward trend of atmospheric Hg, we give the emission inventory (Table S6) and concentrations (Table S5) of other air pollutants in the studied regions in both 2014 and 2016. Both the emissions and concentrations of $SO_2$, $NO_2$, and PM showed a decreasing trend, which is used to support that "air pollution control policies targeting $SO_2$,

$NO_2$, and PM reductions had significant co-benefits on atmospheric Hg".

"Table 3 showed the detailed data of the three classifications. From 2014 to 2016, the whole

China region (NCP, SW-YRD) contributed to 70% of GEM decline at Chongming Island.

Considering downward trend of emission inventory and atmospheric pollutant from 2014 to

2016 in NCP and SW-YRD region (Table S5, Table S6), the reason of downward trend can be attributed to the effectiveness of existing air pollution control measures in China (SC, 2013;

MEP, 2014)."

**See the revised manuscript at line 380 – 384**

**Table S5.** The annual concentration of $SO_2$, $NO_x$, $O_3$ and $PM_{2.5}$ at Chongming site, NCP, and

SW-YRD regions

| Year | | 2014 | | | 2016 | | | Change | | |
|---|---|---|---|---|---|---|---|---|---|---|
| Pollutants | Region | NCP | SW-YRD | Chongming | NCP | SW-YRD | Chongming | NCP | SW-YRD | Chongming |
| $PM_{2.5}$ ($\mu g\ m^{-3}$) | | 71.93 | 53.05 | 25.09 | 60.75 | 44.75 | 23.89 | -16% | -16% | -5% |
| $SO_2$ | | 34.52 | 21.01 | 1.60 | 24.37 | 16.40 | 1.47 | -29% | -22% | -8% |

| | | | | | | | | | |
|---|---|---|---|---|---|---|---|---|---|
| (µg m$^{-3}$) | | | | | | | | | |
| NO$_2$ (µg m$^{-3}$) | 45.07 | 34.34 | 12.62 | 41.55 | 34.40 | 10.84 | -8% | 0% | -14% |
| O$_3$ (µg m$^{-3}$) | 60.29 | 56.27 | 41.70 | 61.84 | 60.92 | 44.38 | 3% | 8% | 6% |
| GEM (ng m$^{-3}$) | No data | | 2.68 | No data | | 1.60 | No data | | -40% |

Note: According to the contribution of trajectory, the dominant provinces in the NCP region
included Beijing, Tianjin, Hebei, Shandong and Liaoning province. The SW-YRD mainly
contained Shanghai, Zhejiang, Jiangsu, Jiangxi and Anhui province.

**Table S6.** Emission inventories of the main pollutants from the studied regions in 2014 and
2016

| Air pollutants | 2014 | | 2016 | | Decline proportion | |
|---|---|---|---|---|---|---|
| | NCP | SW-YRD | NCP | SW-YRD | NCP | SW-YRD |
| PM$_{2.5}$ (kt) | 2019 | 1209 | 1849 | 1109 | -8% | -8% |
| NO$_x$ (kt) | 5697 | 4022 | 5424 | 3855 | -5% | -4% |
| SO$_2$ (kt) | 3780 | 1993 | 3450 | 1780 | -9% | -11% |
| GEM (t) | 118 | 72 | 103 | 67 | -13% | -7% |

Note: According to the contribution of trajectory, the dominant provinces in the NCP region
included Beijing, Tianjin, Hebei, Shandong and Liaoning province. The SW-YRD mainly
contained Shanghai, Zhejiang, Jiangsu, Jiangxi and Anhui province.

*Comment 9:*

Lines 173-175: "Besides, this method required similar meteorological conditions of the periods participated in comparison so as to reduce the interference from meteorology". I am not sure I

understand this sentence. Do you mean that you used similar meteorological data in 2014 and

2016 to compute the back-trajectories? Or are you referring to the fact that meteorological conditions were pretty much similar in 2014 and 2016 (lines 266-274)?

*Response:*

Thanks for the comments. Yes, this sentence is referring to the fact that meteorological conditions were pretty similar in 2014 and 2016. We have revised the sentence as suggested to make it easier to understand.

"Besides, meteorological conditions were pretty similar in 2014 and 2016 so as to reduce the interference from meteorology (Table S2).

**See the revised manuscript at line 203 – 204**

*Comment 10:*

Lines 188: "For small emission sectors (: : :)". Which ones?

*Response:*

   We have added the explanation of small emission sectors in the revised manuscript as below.

   "The emission sectors included coal-fired power plants, coal-fired industrial boilers, residential coal-combustion, cement clinker production, iron and steel production, mobile oil combustion, and other small emission sectors (eg., zinc smelting, lead smelting, municipal solid incineration, copper smelting, aluminum production, gold production, other coal combustion, stationary oil combustion, and cremation)."

   **See the revised manuscript at line 212-216.**

*Comment 11:*

Lines 193-194: "The average concentrations of GEM in 2014 and 2016 were (: : :)". What about the mean concentration in 2015? Additionally, are the average annual concentrations actually referring to March-December? If so, please add something like "The average concentrations of GEM in 2014 (Mar-Dec) and 2016 (Mar-Dec) were (: : :)".

*Response:*

   The average concentration of GEM in 2015 was $2.14 \pm 0.82$ ng m$^{-3}$. And we have added the concentration of 2015 and remark in the revised manuscript.

   "The average concentrations of GEM in 2014 (March to December), 2015 and 2016 (March to December) were $2.68 \pm 1.07$ ng m$^{-3}$, $2.14 \pm 0.82$ ng m$^{-3}$, and $1.60 \pm 0.56$ ng m$^{-3}$, respectively."

   **See the revised manuscript at line 230 – 231**

*Comment 12:*

Lines 194-195: How does it compare to concentrations reported in Sprovieri et al. (2016)?

*Response:*

   $t$ test was used to compare the GEM concentration at Chongming and background concentration of Northern Hemisphere. The $p$ value ($p<0.01$) of the $t$ test were added in the revised manuscript as below.

"The GEM concentrations in 2014 (2.68±1.07 ng m$^{-3}$) were higher ($t$ test, $p<0.01$) than the

Northern Hemisphere back-ground concentration (about 1.5 ng m$^{-3}$) (Sprovieri et al., 2010) and those measured in other remote and rural locations in China (Zhang H et al., 2015; Fu et al.,

2008a; Fu et al., 2009)."

**See the revised manuscript, line 231 – 234**

*Comment 13:*

Lines 199-200: "During this period, monthly GEM concentrations showed a significant decrease with a rate of -0.60 ng/m3/yr". Table 1 refers to TGM concentrations, not GEM.

Additionally, as mentioned earlier, the rate is -0.52 ng/m3/yr in Table 1. Please, try to be consistent throughout the manuscript.

*Response:*

Thanks for the comments. It is a typo. We have gone through the whole paper so as to make the manuscript consistent.

*Comment 14:*

Lines 201-216: To me, "GEM" and "TGM" are not interchangeable (see previous comment).

While the difference between TGM and GEM is usually smaller than 1% (Soerensen et al.,

2010), it might not be the case everywhere. What is the fraction of GOM at your site? I would appreciate a discussion on analytical uncertainties and instrumental setups.

*Response:*

We agree that the GEM and TGM are not always interchangeable. The average concentration of GOM during sampling period was 14.81 ± 13.21 pg m$^{-3}$, which was less than 1% of TGM.

Thus, the GEM concentrations were approximated to TGM concentrations July 5, 2015 to April

30 2016 when the speciation unit does not work, as most of other studies have done (Slemr et al., 2015; Sprovieri et al., 2016). We have pointed out this in the revised manuscript.

A discussion on analytical uncertainties and instrumental setups has been added in the following text as below.

"From July 5, 2015 to April 30 2016, the Tekran 1130/1135 speciation unit was damaged by the rainstorm, the Tekran 2537X were operated without speciation units but with PTFE filter to protect the instrument from particles and sea salt. Therefore, the observed concentrations during

July 2015-April 2016 were TGM concentrations indeed. However, the GOM concentrations at

Chongming Island accounted for less than 1% of TGM (TGM=GOM+GEM). Thus, the GEM

concentrations were approximated to TGM concentrations from July 2015 to April 2016."

**See the revised manuscript at line 109 – 115**

"In our research, random uncertainties of individual measurement had been averaged out and the systematic uncertainties need to be considered. The overall practically achievable systematic uncertainty would be 10% considering that the instrument was not in ideal performance (Slemr et al., 2015; Steffen et al., 2012). For example, slow deactivation of the traps, contamination of the switching valves and leaks would increase the uncertainties but were difficult to quantify (Slemr et al., 2015;Steffen et al., 2012). Because of the consistency of instrument and the quality assurance/quality control have been paid special attention to during the sampling campaign, the systematic differences of instrument did not affect the huge variation between 2014 and 2016."

**See the revised manuscript at line 124 – 131**

The sentence "at the Cape Point of South Africa, GEM concentrations decreased from 1.35

ng/m3 in 1996 to 0.9 ng/m3 in 2008" is not entirely true. A downward trend has been observed from 1996 to 2005, while an upward one is observed since 2007 (Martin et al., 2017; Slemr et al., 2015).

*Response:*

We have revised our expression about observation trend in Cape Point as follows

"In South Africa, annual average GEM concentration at Cape Point decreased from 1.29 ng m$^{-3}$

in 1996 to 1.19 ng m$^{-3}$ in 2004 (Slemr et al., 2008) and were increasing from 0.93 ng m$^{-3}$ in

2007 (Slemr et al., 2015) until 2016 (Martin et al, 2017)."

Additionally, the instrumental setup changed: a manual amalgamation technique was used from

1995 to 2004 while a Tekran instrument has been used since 2007 (Martin et al., 2017). It might also be the case at other stations in Table 1. How does it influence the various trends reported in Table 1?

***Response:***

Table 1 was moved to supporting information (Table S4) so as to focus on our topic. In the

Table S4, all the stations used Tekran instruments except for the monitoring in South Korea.

The instrument of Canadian sites were maintended by the Environment Canada- developed

Research Data Management and Quality Assurance System (RDMQ). At Zeppelin, the instreument maintenance were carried out by the protocols of Norwegian mercury monitoring program. Instruments at these plances have been maintended under the guidance of similar quality control criteria. In South Korea, the concentration of TGM were measured using an automatic on-line analytical system called a AM-series analyzer. Although the intruments used in the stations listed in the Table 1 were not totally the same, the instruments at each site remained unchanged during the monitoring period. Therefore, the downward trend at different sites can be compared in the Table S4. We have revised the Table S4 and give expression about the different instrument in the revised manusript.

"All the stations in Table S4 used Tekran instruments except for the observation in South

Korea. Different instruments could cause potential differences in the observation, but they were comparable and did not affect the conclusion of comparison in downward trend (Slemr et al.,

2015; Sprovieri et al., 2016)."

**See the revised manuscript at line 253 - 256**

**Table S4.** Historical variation trends of atmospheric Hg in previous studies

| Monitoring site | Duration | TGM trend ($pg\ m^{-3}\ yr^{-1}$) | Variation trend | Site description | Monitoring instrument | References |
|---|---|---|---|---|---|---|
| Alert, Canada | 2000-2009 | -13(-21,0) | -0.9% $y^{-1}$ | Remote | 2537A | Cole et al. 2013 |
| Kuujjuarapik, Canada | 2000-2009 | -33(-50,-18) | -2.1% $y^{-1}$ | Remote | 2537A | Cole et al. 2013 |
| Egbert, Canada | 2000-2009 | -35(-44,-27) | -2.2% $y^{-1}$ | Remote | 2537A | Cole et al. 2013 |
| Zeppelin Stn, Norway | 2000-2009 | +2(-7,+12) | - | Remote | 2537A | Cole et al. 2013 |
| St.Anicet, Canada | 2000-2009 | -29(-31,-27) | -1.9% $y^{-1}$ | Remote | 2537A | Cole et al. 2013 |
| Kejimkujik, Canada | 2000-2009 | -23(-33,-13) | -1.6% $y^{-1}$ | Remote | 2537A | Cole et al. 2013 |

| Site | Period | | | Classification | Instrument | Reference |
|---|---|---|---|---|---|---|
| Head, Ireland | 1996-2009 | - | $-1.3 \pm 0.2\%$ y$^{-1}$ | Rural | 2537A | Weigelt et al. 2015 |
| Yong San, South Korea | 2004-2011 | No trend (3.54±1.46 ng m$^3$) | | Urban | AM-3 | Kim et al. 2016 |
| Yong San South Korea | 2013-2014 | Decrease to 2.34±0.73 ng m$^3$ | | | AM-3 | Kim et al. 2016 |
| Mt. Changbai | 2013-2015 | Decrease from 1.74 ng m$^{-3}$ to 1.58 ng m$^{-3}$ | | Remote | 2537B | Fu et al. 2015 |
| Chongming Island, China | 2014-2016 | -600 | -29.4%/y | Remote | 2537X | This study |

*Comment 15:*

Lines 212-214: "The decreasing trend observed in our study was accordant with the unpublished data in Mt. Changbai during 2014-2015 cited in the review of Fu et al. (2015). But much sharper decrease of Hg concentrations was observed in our study". Aren't the data at Mt. Changbai you are referring to in Sprovieri et al. (2016)? What is the trend at that site? Why isn't included in Table 1?

*Response:*

The data of Mt. Changbai reported by Sprovieri et al. (2016) is the observation in 2013. The observation period is not in the range of our study period. Therefore, we cited the data reported by Fu et al.(2015), where the observation is in 2014-2015.

"The decreasing trend observed in our study was accordant with the data in Mt. Changbai during 2014-2015 cited in the review of Fu et al. (2015). The atmospheric mercury at Chongming was influenced by and in turn reflected regional mercury emission and cycle. Although the decline in atmospheric mercury was observed in many sites of the Northern Hemisphere, much sharper decrease of Hg concentrations was observed in our study."

**See the revised manuscript at line 266 - 271**

*Comment 16:*

Line 224: Are you referring to Figure 2?

*Response:*

Refer to figure 3 in our revised manuscript. We have corrected this in the revised manuscript.

***Comment 17:***

Lines 225-227: Is that based on the _3 years of data?

***Response:***

Figure 3 is calculated based on the 3 years data. We have revised it in the manuscript as below.

"According to the decomposition result (Figure 2c), we observed strong seasonal cycle with seasonal GEM peak in July and trough in September, so GEM concentrations in the same month but different years were averaged to discuss the seasonal circle (Figure 3)."

**See the revised manuscript at line 274 - 276**

***Comment 18:***

Line 234: "The higher Hg concentrations in cold seasons in Mt. Ailao and Mt. Waliguan (: : :)".

You say above that concentrations are lower in the cold season at these sites. This is confusing.

***Response:***

Sorry for the mistakes. We have revised the manuscript as below.

"The higher Hg concentrations in cold seasons in Mt. Leigong were mainly explained by coal- combustion for urban and residential heating during cold seasons. Whereas, increasing solar radiation and soil/air temperature dominate the higher Hg concentrations in Mt. Ailao."

**See the revised manuscript at line 288- 291**

***Comment 19:***

Line 250-251: "Therefore, we supposed that the seasonal cycle of GEM concentrations was dominated by natural emissions". How can you explain that the seasonal cycle is more pronounced in 2014 than in 2016? See major comment.

***Response:***

The seasonal variation was more pronounced in 2014 can attribute to the lower wet deposition and GEM oxidation. On one aspect, as a costal site, the Chongming island is abundant with •OH. The increase of $O_3$ concentration from the summer of 2014 to 2016 may contribute to a higher oxidation of GEM in 2016. On another aspect, and higher wet Hg deposition is approximately 6.6 times of that in the winter at Chongming (Zhang et al., 2010).

Meanwhile, the rainfall in 2016 summer (546 mm) was higher than the rainfall in 2014 (426

mm). Therefore, the higher oxidation and wet deposition rate of mercury in the summer of 2016

will reduce the concentration difference between summer and winter, which lead to a less pronounced seasonal variation in 2016.

[Figure]

**Figure 2.** Monthly average GEM concentrations during the studied period (a) observed monthly

GEM concentrations (b) GEM trend after decomposition (c) GEM seasonality after decomposition (d) GEM random after decomposition

Note: The observed concentrations during July 2015-April 2016 were TGM concentrations indeed due to the problems of Tekran 1130/1135. However, the GOM concentrations at

Chongming island accounted for less than 1% of TGM. Thus, the GEM concentrations were approximated to TGM concentrations during July 2015-April 2016.

"From Figure 1, we also observed more pronounced seasonal variation in 2014, which can be attributed to the lower wet deposition and GEM oxidation. On one aspect, as a costal site, the Chongming Island is abundant with •OH. The increase of $O_3$ concentration from the summer of 2014 to 2016 may contribute to a higher oxidation of GEM in 2016. On another aspect, and higher wet Hg deposition is approximately 6.6 times of that in the winter at Chongming (Zhang et al., 2010). Meanwhile, the rainfall in 2016 summer (546 mm) was higher than the rainfall in

2014 (426 mm). Therefore, the higher oxidation and wet deposition rate of Hg in the summer of 2016 will reduce the concentration difference between summer and winter, which lead to a less pronounced seasonal variation in 2016. Meanwhile, the higher oxidation and wet deposition in 2016 also contributed to the downward trend of GEM by reducing the seasonality in spring and summer (Figure S3)."

**See the revised manuscript at line 318 - 328**

***Comment 20:***

Lines 275-276: "This decline may be contributed by the downward trend of GEM

concentrations in north hemisphere". Please, elaborate on this idea. I don't really understand what you mean here.

***Response:***

Sorry for the obscure expression. The decline of PSCF value means that East China Sea has less contribution to Chongming in 2016. The potential reason of the decline on PSCF value in the East China Sea may be attributed to the reduction of emissions in Japan and Korea. The downward trend in Japan and Korea will lead to clean air mass transport from Japan and Koran to East China Sea and then to Chongming. We have revised the manuscript as below:

"The decline from the East China Sea may be contributed by the downward trend of GEM

concentrations in South Korea and Japan (Kim et al., 2016; Kim et al., 2013), where the anthropogenic Hg emissions of Japan and South Korea have been reduced by 13% and 4%

during 2010-2015, respectively (UNEP 2013; UNEP 2018). The air mass from Japan and South

Korea would pass through the East China Sea to Chongming."

**Table R2.** Total Hg emission from Japan and South Korea in 2010 and 2015

| Country | Mercury emissions (t) | | Decline | Reference |
| --- | --- | --- | --- | --- |
| | 2010 | 2015 | | |
| Japan | 17.07 | 14.86 | -13% | UNEP Technical Report (2013) |
| South Korea | 7.32 | 7.01 | -4% | UNEP Technical Report (2018) |

**See the revised the manuscript at line 336-340**

*Comment 21:*

Lines 315-325: Do you get the same results if you perform this analysis on SO2, NOx, and PM

concentrations?

*Response:*

The $SO_2$, $NO_x$ and $PM_{2.5}$ concentrations at Chongming island also show downward trend.

However, such kind of analysis is not so suitable for SO2, NOx, and $PM_{2.5}$ due to the following reasons. First, the residential time of $SO_2$, NOx, and $PM_{2.5}$ is 2-4 d, 8-10d, and several days to few weeks, respectively (Pirrone, et al., 1996, Seinfeld, Spyros, 2016). Such residential time is much shorter than that for $Hg^0$. Second, SO2, $NO_x$ and $PM_{2.5}$ are more reactive in the atmosphere compared with $Hg^0$ (Pirrone, et al., 1996, Seinfeld, Spyros, 2016).

*Comment 22:*

Line 318: 34% should be 35% according to Table 4. Additionally, how can you explain this result? Is there a decline in anthropogenic emissions and a GEM decreasing trend in this region (China Sea, Japan, South Korea) as well? Cluster EAST explains 35% of the decline, i.e., 0.35

x 0.52 = 0.182 ng/m3/yr. Is that consistent with trends reported in this region (e.g., Kim et al.,

2016)?

*Response:*

Yes. This is a mistake that 34% should be 35% in the original manuscript.

However, we have changed the definition of cluster according to the suggestion. The NCP

region, SW-YRD region, and ABROAD region causes 26%, 44%, and 30% for GEM decline, respectively. The whole China region (NCP, SW-YRD) contributed to 70% of GEM decline at

Chongming Island while ABROAD region contributed to 30%. The decline in NCP and SW-

YRD indicated effective air pollution control policy in China since 2013. The decline in

ABROAD region was originated from GEM decline in South Korea and Japan.

The decline in Chongming was consistent with the decline in anthropogenic emission and a

GEM decreasing trend in the ABROAD region. In South Korea, the decline of GEM at Seoul can be calculated as 0.47 ng m$^{-3}$ yr$^{-1}$ from 2011 to 2013 (Kim, et al., 2016, Kim, et al., 2013).

In Japan, there is no published data about long term trend since 2010. But the emission inventory of Japan decreased from 2010 to 2015 (Table R2). Therefore, the decline in

ABROAD can be attributed to the decline in South Korea and Japan.

**Table R2.** Total Hg emission from Japan and South Korea in 2010 and 2015

| Country | Mercury emissions (t) | | Decline | Reference |
|---|---|---|---|---|
| | 2010 | 2015 | | |
| Japan | 17.07 | 14.86 | -13% | UNEP Technical Report (2013) |
| South Korea | 7.32 | 7.01 | -4% | UNEP Technical Report (2018) |

We also revised the manuscript as below.

"The decline from the East China Sea may be contributed by the downward trend of GEM

concentrations in South Korea and Japan (Kim et al., 2016; Kim et al., 2013), where the anthropogenic Hg emissions of Japan and South Korea have been reduced by 13% and 4%

during 2010-2015, respectively (UNEP 2013; UNEP 2018). The air mass from Japan and South

Korea would pass through the East China Sea to Chongming."

**See the revised manuscript at line 336 -340**

*Comment 23:*

Lines 321-323: "We also noted that the largest decline of Hg concentrations was observed in the cluster SW, which indicated more effective air pollution control in the regions where the air mass of the cluster SW passed". What about the seasonality of GEM concentrations in the various clusters (NW, SW and EAST)? Could a difference in seasonality explain the observed

Hg decline?

*Response:*

The seasonality of GEM concentration in the various clusters was showed in Figure S3. In our revised manuscript, cluster NW, SW and EAST were modified to cluster NCP, SW-YRD

and ABROAD. The seasonality of cluster NCP, SW-YRD and ABROAD were similar to the seasonality at Chongming. The GEM concentration of different clusters reached the highest in the summer of 2014. And the seasonality in 2014 for the three clusters was more pronounced than their seasonality in 2016.

The seasonality also explained the observed decline. From 2014 to 2016, all the clusters declined in all season. In 2014, the seasonality was more pronounced than the seasonality in

2016. It can be attributed to the higher oxidation of GEM and higher wet deposition in 2016.

The smaller seasonal variation also had an effect on the decline. We revised our expression in the revised manuscript.

"Therefore, the higher oxidation and wet deposition rate of mercury in the summer of 2016

will reduce the concentration difference between summer and winter, which lead to a less pronounced seasonal variation in 2016. Meanwhile, the higher oxidation and wet deposition in

2016 also contributed to the downward trend by reducing the seasonality of spring and summer (Figure S3)."

**See the revised manuscript in line 324 – 328.**

[Figure]

**Figure S3.** The seasonality of GEM concentration in the NCP, SW-YRD and ABROAD (No trajectory transport though ABROAD in winter of 2014)

**See the revised manuscript at Figure S3.**

*Comment 24:*

Figure 3: Could you please add the standard deviations? Is that the average over several years?

*Response:*

We have revised as suggested. It is the average in the three years. We also give expression in the manuscript as below.

"According to the decomposition result (Figure 2c), we observed strong seasonal cycle with seasonal GEM peak in July and trough in September, so GEM concentrations in the same month but different years were averaged to discuss the seasonal circle (Figure 3). The average data can eliminate the effect of downward trend and get result of average seasonal variation. The error bars in the Figure 3 mean the standard deviation of the monthly average."

**See the revised manuscript in line 274– 278.**

[revised manuscript text omitted]